



# A Novel Simplified Ground-Based TIR System for Volcanic Plume Geometry, SO₂ Columnar Abundance, and Flux Retrievals

Lorenzo Guerrieri[1], Stefano Corradini[1], Luca Merucci[1], Dario Stelitano[1], Fred Prata[2], Linda Lambertucci[1], Camilo Naranjo[1], Riccardo Biondi[3]

[1]Istituto Nazionale di Geofisica e Vulcanologia, Via di Vigna Murata, 605, 00143 Roma, Italy
[2]AIRES Pty Ltd., Melbourne, VIC 3930, Australia
[3]CIMA Research Foundation, Via A. Magilotto, 2, 17100 Savona, Italy

*Correspondence to*: Lorenzo Guerrieri (lorenzo.guerrieri@ingv.it)

**Abstract.** In the last few decades, volcanic monitoring using remote sensing systems has become an essential tool to investigate the effects of volcanic activity on environment, climate, human health and aviation, as well as to give insights into volcanic processes. Compared to satellite measurements, ground-based instruments offer continuous spatial and temporal coverage capable of providing high resolution and high sensitivity data.

This work presents a new simplified prototype of a Thermal InfraRed (TIR) system (named "VIRSO2"). The instrument comprises three cameras, one working in the visible and two in the TIR (8-14 μm). In front of one of the two TIR cameras, an 8.7 μm filter is placed. The system is designed for detection of volcanic emission, geometry estimation, columnar content of $SO_2$ and ash, and $SO_2$ flux retrievals. The retrieval procedures developed are detailed starting from the geometric characterization with wind direction correction, the calibration by considering the effects of filter multireflections and temperature, and the $SO_2$ mass by exploiting MODTRAN radiative transfer model (RTM) simulations. The $SO_2$ flux is then computed by applying the traverse method, with the plume speed obtained from the wind speed at the crater altitude. As test cases, the measurements collected at Etna volcano (Italy) on the 1 April 2021 during a lava fountain episode and the 30 August 2024 during a quiescent phase have been considered. The results show that the system can provide reliable information on plume detection, altitude, and $SO_2$ flux.

The simplicity, low cost, and the possibility of carrying out measurements at a safe distance from the vent both day and night, make this system ideal for real-time monitoring of volcanic emissions, thus helping to provide information on the state of activity of the volcano and therefore to mitigate the effect that these natural phenomena have on humans and the environment.

## 1 Introduction

During their activities, volcanoes emit large amounts of particles (ash, water droplets) and gases (mostly $H_2O$, $CO_2$, $SO_2$ and HCl) into the atmosphere, with severe threats on the environment (Thordarsson et al., 2003; Craig et al., 2016), climate (Robock, 2000; Haiwood et al., 2000; Grainger et al., 2003; Solomon et al., 2011; Savigny et al., 2020), human health (Horwell et al., 2006; 2013) and aviation (Casadevall et al., 1994; Kristiansen et al., 2015). In the last few decades, ground and satellite remote sensing systems have become essential tools for monitoring volcanic activity in real-time to mitigate the effect of these natural phenomena and give insight into volcanic processes. If satellite measurements guarantee global spatial coverage with generally low spatial resolution and variable accuracy, ground systems offer continuous spatial and temporal coverage capable of providing high resolution and sensitivity but are punctual.

From satellite, the volcanic clouds are generally detected in the Thermal InfraRed (TIR) spectral range by exploiting the opposite spectral behaviour between ash and water particles around 11 μm and 12 μm (Prata et al., 1989; Wen & Rose,



1997) and the wide $SO_2$ absorption features at 7.3 μm and 8.7 μm (Realmuto et al., 1994; Watson et al., 2004). The absorption feature at 7.3 μm is strongly affected by the atmospheric water vapour, so it is generally used when a volcanic cloud is located at high altitudes, while 8.7 μm lies in an atmospheric window and then can be exploited for any volcanic cloud altitude (Watson et al., 2004; Corradini et al., 2009). Moreover, remote sensing in the TIR guarantees the possibility to realise monitoring during both day and night.

As emphasised by Prata et al., 2024, the TIR ground-based remote sensing applied to studies on volcanic plumes, is a relatively new application and had a strong increase in use following the wide availability of uncooled thermal cameras. These systems have been used to characterise volcanic activity at Stromboli (Italy) (Ripepe at al., 2004; Patrick et al., 2007; Patrick, 2007), to investigate unsteadiness in low-energy eruptions of Sabancaya Volcano (Peru) (Rowell et al., 2023), to study the dynamics of volcanic plumes at Santiaguito Volcano, Guatemala (Sahetapy-Engel and Harris, 2009),

to track volcanic explosions at Reventador volcano, Ecuador (Vasconez et al., 2002) and as part of a continuous and real-time volcanic monitoring system (Mereu et al., 2023).

In this work, we will describe a system that represents an evolution of the TIR instrument described in Prata et al., 2024. In this case, two TIR and one visible (VIS) cameras are present, and in front of one of the two twin TIR cameras, an 8.7 μm filter is placed. The data collected and properly calibrated, are used to detect volcanic plumes, estimate their altitude

and thickness, and $SO_2$ slant column densities (*SCD*). The $SO_2$ flux is also obtained by knowing the viewing geometry and by exploiting the plume speed computed from European Centre for Medium-Range Weather Forecasts (ECMWF) Reanalysis v5 (ERA5) hourly dataset (Hersbach et al., 2023). As test cases, the measurements collected on Etna (Sicily, Italy) in the period March-April 2021 and August 2024 have been considered. With its current height of 3403 m asl (September 2024), Etna is the largest European volcano. Its intense and continuous activity (e.g. 66 lava fountains from

late 2020 to early 2022) (Guerrieri et al., 2023) makes it the most active volcano in Europe and one of the most active in the world.

The paper is organised as follows: in Section 2 the characteristics of the TIR ground-based system are described. Sections 3, 4, and 5 present the algorithms for the camera view geometrical computation, the calibration, and the $SO_2$ retrieval procedures. The $SO_2$ flux computation with validation and sensitivity study is described in Sections 6 and 7, while the

conclusions are drafted in Section 8.

## 2 System Description

A novel TIR ground-based camera system (called "*VIRSO2*") has been developed in the sphere of the ESA-VISTA project (https://eo4society.esa.int/projects/vista/, last access: 7 January 2025) by AIRES Pty Ltd company (https://www.aires.space/, last access: 7 January 2025). The system consists of 3 cameras (see Figure 1a): one VIS, one

TIR broadband (7.8-14 μm, BB-broadband), and another similar TIR broadband with a narrowband filter in front (centred at 8.7 μm, NB-narrowband) to exploit the $SO_2$ strong absorption signature. Figure 1b shows a table with its main characteristics: the two TIR broadband cameras consist of a 320 × 240 uncooled microbolometer detector array (manufactured by SEEK Thermal, MOSAIC Core C3 Series https://www.thermal.com/mosaic-core-320x240-4mm.html, last access: 7 January 2025). Figure 1c shows the spectral response functions (SRFs) of BB (blue line) and NB (red line)

respectively used in the MODerate resolution atmospheric TRANsmission (MODTRAN) 5.3 (Berk et al., 2004) simulations needed for quantitative retrievals. As the figure shows, a simple top-hat function between 7.8 μm and 14 μm was considered for BB since there is no specific information about it from SEEK Thermal. For NB, the manufacturer supplied spectral transmittance (normalised to 1) of the 8.7 μm filter was used. The MODTRAN spectral radiances were weighted by the two SRFs and then converted into brightness temperatures by inverting the Planck function considering

a central wavelength of 998 cm$^{-1}$ (10.02 μm) and 1151 cm$^{-1}$ (8.69 μm) respectively.



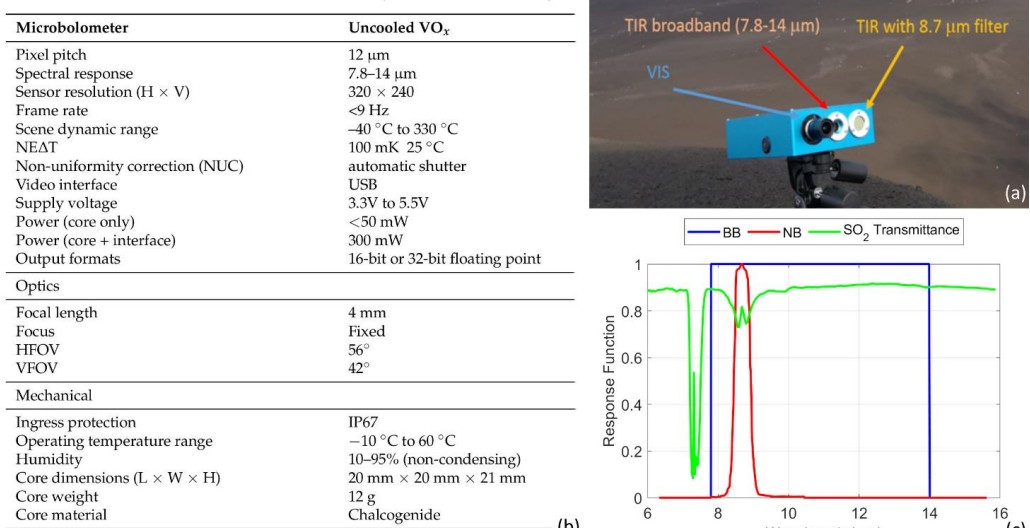

**Figure 1. (a) View of the system with the detail of VIS, TIR cameras and 8.7 µm filter (b) Specifications for the SEEK Thermal IR camera (MOSAIC Core C3 Series), taken from Prata et al., 2024. (c) Spectral response functions used for broadband (BB) and narrowband (NB) channels with SO₂ transmittance spectrum.**

The system is controlled by a Single-Board Computer with 500 GB memory for data storage, complete with wi-fi, Bluetooth, dual ethernet ports, USB, HDMI, and a microSD card slot to ensure adequate connectivity under most conditions. The system is operated using an external power bank. It is extremely portable (total weight of about 3 kg) and relatively low cost (about 9 kEuros). Its ability to be used both day and night and its good sensitivity makes it a very important tool for the continuous and real-time monitoring of volcanic emissions from the ground. More information about the system can be found in Prata et al. (2024). Figure 2 shows a schematic flowchart of the procedure for $SO_2$ retrieval (a) and an example of a BB temperature image collected at Etna volcano on 31 March 2021, during an eruptive event (b). The maximum value of the colormap is set to 300 K to make more evident the volcanic plume, but actually the maximum value detected by the camera is 443 K. The minimum value, relating to clear sky, is instead 233 K.

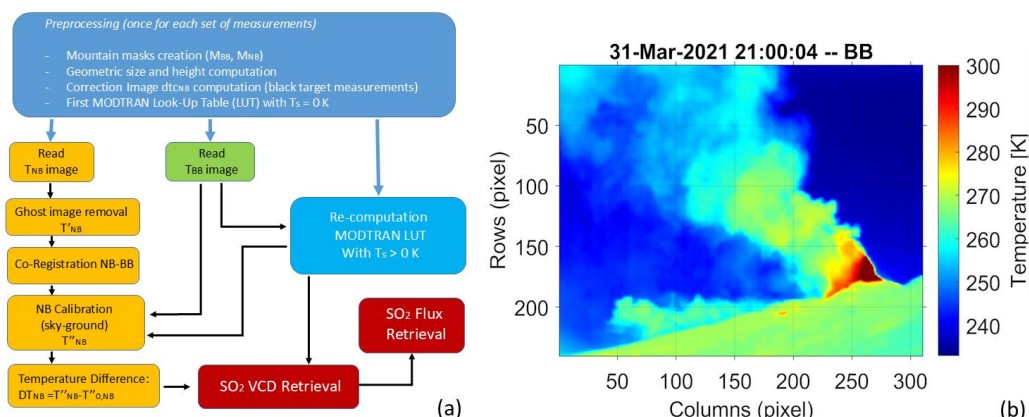

**Figure 2: (a) Flowchart of the developed procedure for SO₂ retrieval. (b) VIRSO2 TIR broadband (BB) image collected at Etna (top of the "Sapienza" cableway, 2500 m asl) the 31 March 2021 at 21:00:04 UTC.**





### 3 Camera Geometry Computation

The knowledge of the camera pixel dimensions is fundamental for the plume geometry (altitude and thickness) retrievals and $SO_2$ total mass or flux computations. Pixel size depends both on the focal length and the Field Of View (FOV) of the detector, as well as on the distance between the system and the target and the inclination of the system itself. The procedure is based on geometrical considerations with the main assumption that the focal plane is perpendicular to the ground. This approximation is generally adopted for analysis of ascending volcanic plume in image data (Valade et al., 2014; Bombrun et al., 2018; Simionato et al., 2022). The wind orientation is an important factor to take into account in case of volcanic plumes, so it was implemented in the procedure (see Sect. 3.1).

As summarised in Fig. 1b, the TIR cameras are composed of 320 columns ($NC$) and 240 rows ($NR$), with a Horizontal and Vertical Field Of View ($HFOV$ and $VFOV$) of 56° and 42° respectively. Figure 3a shows the camera measurement configuration considered. Assuming the elevation angle of the camera ($\alpha$) referred to the centre of the image, it is possible to obtain the angular position ($\theta, \varphi$) of the different grid points:

$$\theta(i) = \alpha + \left(\frac{NR}{2} + 1 - i\right) \cdot \frac{VFOV}{NR} \tag{1}$$

$$\varphi(j) = \left(j - \frac{NC}{2} - 1\right) \cdot \frac{HFOV}{NC} \tag{2}$$

where $i = 1, NR + 1$ (from top to bottom) and $j = 1, NC + 1$ (from left to right). In this way, $\theta(i)$ and $\varphi(j)$ range between ($\alpha$+21°, $\alpha$-21°) and (-28°, +28°) respectively.

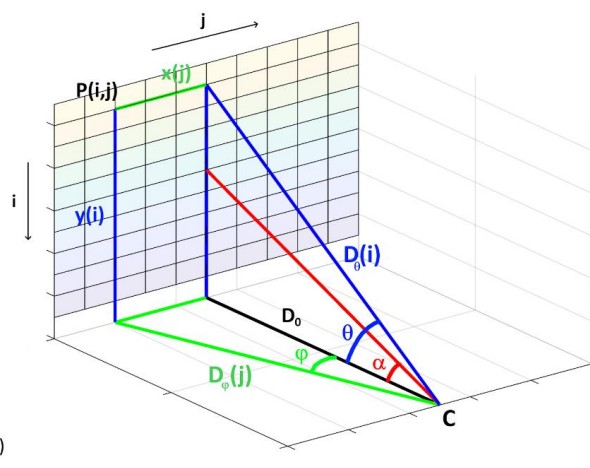

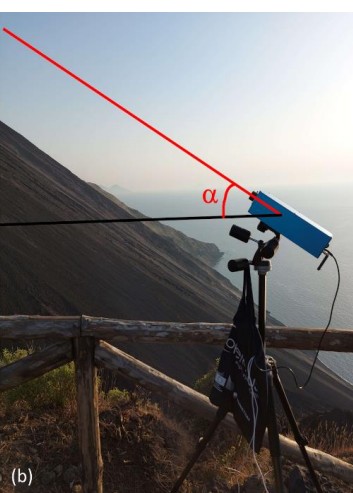

(a)  (b)

**Figure 3: (a) Schematic geometrical configuration of VIRSO2 camera FOV. The camera is placed in C. (b) VIRSO2 camera at Stromboli (Italy). (Note that Earth curvature effects are ignored as generally the system operates on small scales ~10 km or so).**

The horizontal distance from the central column ($x$) and the vertical distance from the ground ($y$) of each grid point can be obtained considering $D_0$ (perpendicular distance between the camera and the image plane):

$$x(j) = D_0 \cdot tan\big(\varphi(j)\big) \tag{3}$$

$$y(i) = D_0 \cdot tan\big(\theta(i)\big) \tag{4}$$





From the differences between adjacent grid points, the horizontal $(dx)$ and vertical $(dy)$ size, the area $(a)$ and the height asl $(h)$ of each pixel can be obtained $(i = 1, NR; j = 1, NC)$:

$$dx(j) = x(j + 1) - x(j) \tag{5}$$
$$dy(i) = y(i) - y(i + 1) \tag{6}$$
$$a(i, j) = dx(j) \cdot dy(i) \tag{7}$$
$$h(i) = y(i) - dy(i)/2 + h_0 \tag{8}$$

where $h_0$ is the altitude of the location where measurements are performed.

Figure 4 shows the horizontal (x-size, Fig. 4a) and vertical (y-size, Fig. 4b) dimensions of the pixels, the pixel area (Fig. 4c), and the height above sea level (Fig. 4d), computed for the "Etna-Piano del Vescovo" measurements (1 April 2021, Fig. 5).

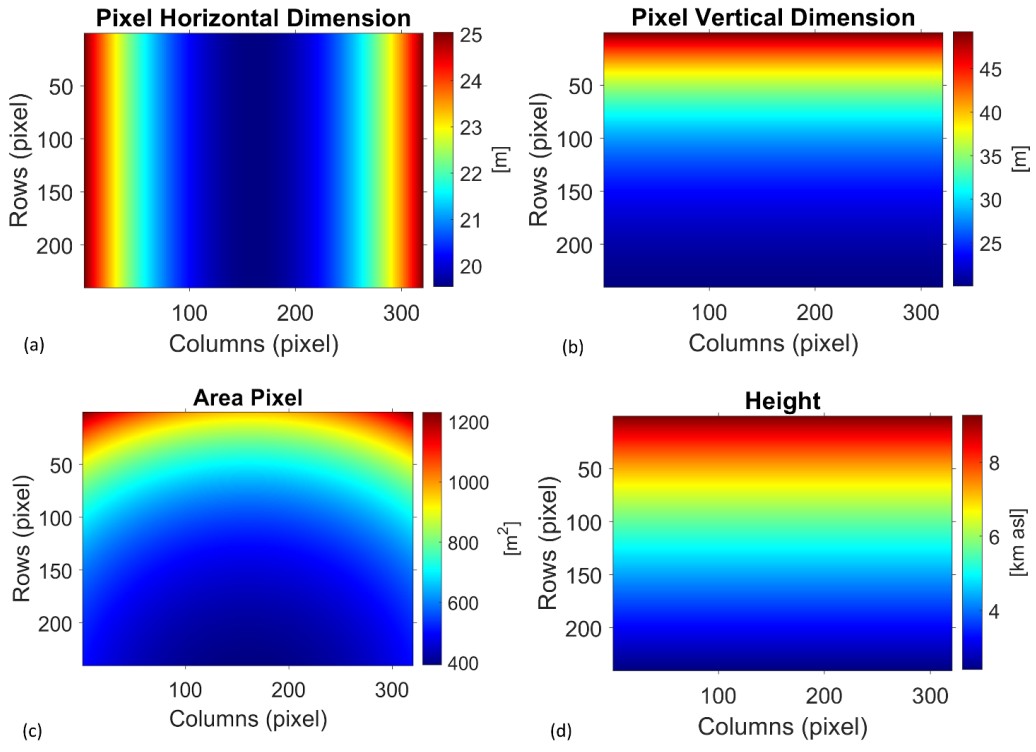

**Figure 4: Pixel dimensions and height obtained with D0 = 6.4 km, α = 30 deg, h0 = 1380 m asl (setting for the camera placed in Piano del Vescovo) and no wind correction. (a) Horizontal pixel dimension dx (m). (b) Vertical pixel dimension dy (m). (c) Pixel area (m2). (d) Height above sea level (km).**

### 3.1 Correction for Wind Direction

The maps obtained in Fig. 4 are correct only if the target is exactly perpendicular to $D_0$, i.e. if the wind direction is parallel to the camera's focal plane. If instead the plume comes towards or goes away from the camera it is necessary to consider





that the horizontal distance from the target changes according to the wind direction and the distance from the crater

position in the camera image (Fig. 5).

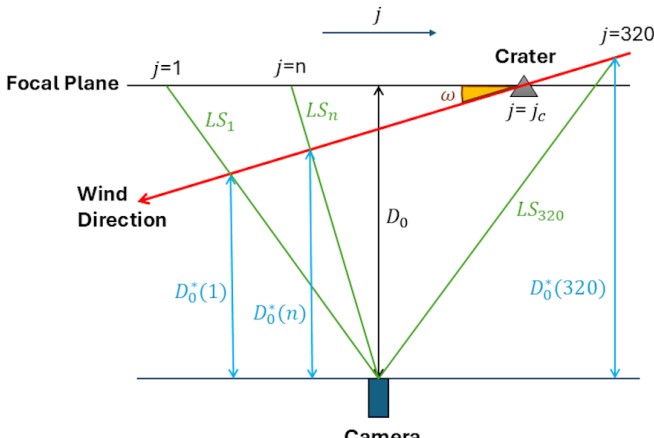

**Figure 5: Bird's-eye schematic view representation to take into account the wind direction (WD, red line) relative to the focal**
**plane of VIRSO2 camera.**

Knowing the column position of the crater ($j_C$) and the relative azimuth angle ($-90° < \omega < 90°$) between the wind

direction (red line) and the focal plane of the camera (black line) it is possible to obtain, for each column of the image,

the point of intersection between the lines of sight (LS) of the camera (green lines) and the wind direction (WD) line. In

a Cartesian reference system in which the camera is placed in the origin and the focal plane is horizontal, the WD and LS

lines are identified by:

$$m_{WD} = tan(\omega) \tag{9}$$

$$q_{WD} = D_0 - m_{WD} \cdot x(j_C) \tag{10}$$

$$m_{LS}(j) = \frac{D_0}{x(j)} \tag{11}$$

$$q_{LS} = 0 \tag{12}$$

where $m_x$ and $q_x$ are the slope and axis intercept respectively of the lines considered.

Finally, the y-position of each of these intersection points gives the new $D_0^*(j)$ (cyan lines in Fig. 5) to be used in eq. 3-8

to compute pixel size dimensions and heights.

Figure 6 shows two BB images of an Etna volcanic plume collected during the paroxysm of 1 April 2021 with and without

the correction for the wind direction. In the upper panels, the eruption was seen during the night from "Piano del Vescovo"

(37.70° N; 15.05° E; 1380 m asl; about 7 km south-east from the crater), while the bottom panels show the volcanic plume

some hours later in the morning from Nicolosi (37.61° N; 15.02° E; 820 m asl; about 15 km south from the crater). The

left panels show the plume heights obtained from eq. 8 without wind correction ($\omega = 0°$), while in the right panels, the

effect of wind correction is considered. In the "Piano del Vescovo" example, $\omega = 26°$ and the top plume height changes

from about 8.5 km to 7 km. In the "Nicolosi" case, where $\omega = 52°$, the improvements are much more important: the top

plume height is now about 9-10 km asl compared to 15-16 km without wind correction. Using the Spinning Enhanced



Visible and InfraRed Imager (SEVIRI) instrument, on board Meteosat Second Generation (MSG) geostationary satellite,

a volcanic plume top altitude of 6.9 km asl was estimated at 00:55 UTC (Guerrieri et al., 2023). This is in good agreement in particular with the "Piano del Vescovo" VIRSO2 nighttime measurements. In the morning, an increase in volcanic activity    produced    a    higher    volcanic    plume,    confirmed    by    INGV    bulletin    n.    258    at    07:42    UTC (https://www.ct.ingv.it/Dati/informative/vulcanico/ComunicatoETNA20210401074037.pdf,    last    access:    7    January 2025), in which it is said that the eruptive cloud exceeded 9 km asl.

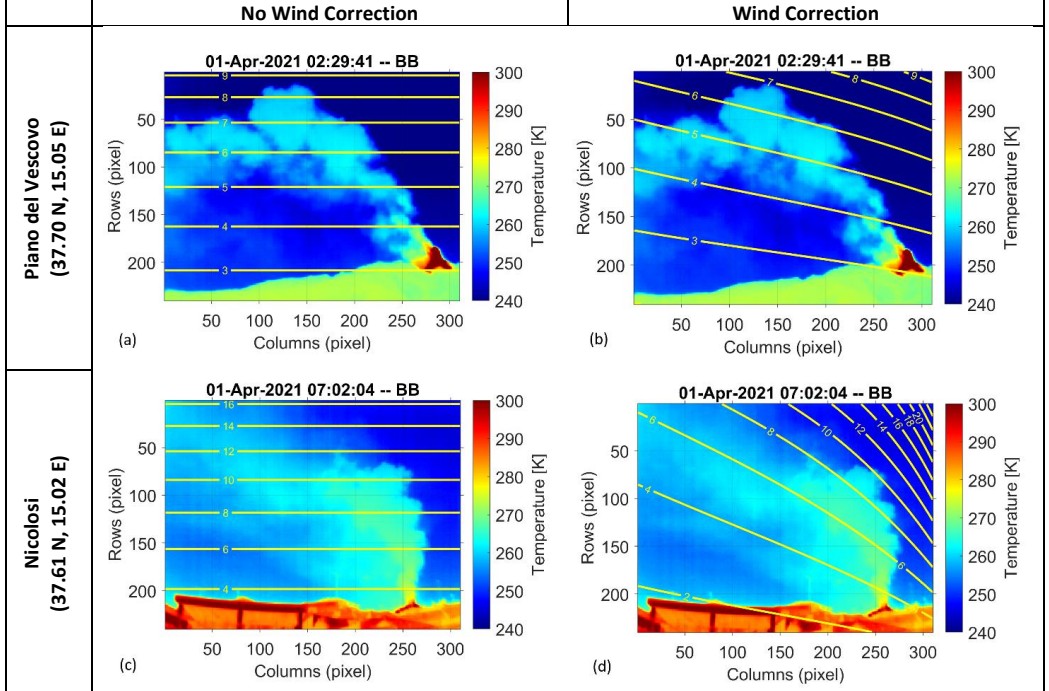

**Figure 6: Top: Etna-Piano del Vescovo (1 April 2021 at 02:29:41 UTC); Bottom: Etna-Nicolosi (1 April 2021 at 07:02:04 UTC). The yellow contour lines (in km) represent the VIRSO2 pixel heights obtained without (left) and with (right) the wind direction correction.**

Table 1 shows the mean (maximum) relative errors of the pixel heights varying $\alpha$ (from 20° to 50°), $D_0$ (from 2.5 to 10 km) and $\omega$ (from 0° to 60°) compared to the calibration tool developed by Snee et al., (2023). The errors increase slightly with the increasing of $\alpha$ and $D_0$, while they increase more rapidly for $|\omega| > 50°$. For wind-focal plane angles $|\omega| < 45°$, the relative errors of all the 320 x 240 pixels of VIRSO2 are always below 1%.

|  |  | $\alpha = 20°$ | $\alpha = 30°$ | $\alpha = 40°$ | $\alpha = 50°$ |
|---|---|---|---|---|---|
| $D_0 = 2.5\ km$ | $\omega = 0°$ | 0.2% (0.2%) | 0.2% (0.2%) | 0.2% (0.2%) | 0.2% (0.4%) |
|  | $\omega = 30°$ | 0.2% (0.2%) | 0.2% (0.2%) | 0.2% (0.3%) | 0.2% (0.5%) |
|  | $\omega = 60°$ | -0.2% (-6.8%) | -0.2% (-7.0%) | -0.2% (-7.1%) | -0.2% (-7.1%) |
| $D_0 = 5\ km$ | $\omega = 0°$ | 0.2% (0.4%) | 0.2% (0.3%) | 0.2% (0.3%) | 0.3% (0.4%) |
|  | $\omega = 30°$ | 0.2% (0.4%) | 0.2% (0.3%) | 0.2% (0.4%) | 0.3% (0.6%) |
|  | $\omega = 60°$ | -0.1% (-7.1%) | -0.2% (-7.1%) | -0.2% (-7.2%) | -0.2% (-7.2%) |
| $D_0 = 10\ km$ | $\omega = 0°$ | 0.4% (0.8%) | 0.3% (0.4%) | 0.3% (0.3%) | 0.3% (0.5%) |



| | | | | | |
|---|---|---|---|---|---|
| | $\omega = 30°$ | 0.4% (0.9%) | 0.3% (0.5%) | 0.3% (0.4%) | 0.3% (0.6%) |
| | $\omega = 60°$ | 0.0% (-7.2%) | -0.1% (-7.2%) | -0.2% (-7.2%) | -0.1% (-7.3%) |

**Table 1: Mean (maximum) relative errors of the pixel heights using the method presented in this paper and the calibration tool of Snee et al. (2023). The altitude of the camera is fixed at 2000 m asl and the crater is positioned in the central column of the VIRSO2 image ($i_C = 200, j_C = 160$).**

## 4 VIRSO2 Calibration

The calibration process is an essential procedure to obtain reliable quantitative results. Here different effects must be
taken into account: at first, the non-perfect transmissivity of the 8.7 μm filter produces a "ghost image". Then, the filter temperature affects the NB measurements and, finally, an adjustment is necessary also for the BB camera, considering that the clear sky temperature often doesn't match with the MODTRAN simulations.

Among the corrections said above, some minor preprocessing tasks are also needed. Firstly, a "mountain mask" is created to distinguish between land and sky pixels. This can be done manually once and for all if the camera is kept in the same
position during the measurement session. Note that two separate mountain masks, one for BB and one for NB, are necessary because the two sensors, even if very close to each other, do not frame exactly the same scene. Given that in the calibration procedure we have to compare BB and NB pixels in the same scene position, the two masks were also used to co-register NB to BB. Finally, a "sky mask" is created too, complementary to the "mountain mask" but excluding also a layer of 15 pixels above the ground to avoid problems with the temperature near the contour of the mountain.

### 4.1 NarrowBand Ghost image removal

All wide-field imagery contains a certain amount of optical vignetting, which is manifested by the gradual dimming of light towards the edges of the image caused by optical components obstructing off-axis light (Haiyan Li and Min Zhu, 2009). For the BB camera, this effect is small enough that a correction is not required (Prata et al., 2024). In addition, a so-called "ghost image" occurs in the NB channel due to the filter placement. It occurs because the interference filter is
not 100% transmissive, which results in a reflection of any hot components behind the filter, and back onto the detector. Since the reflected rays pass back through the focussing optics, an image can be achieved in the center of the detector.

The correction for this effect is obtained by using a purpose-built black target placed in front of the camera. The blackbody plate is large enough to fill the entire FOV and a uniform temperature across the plate is considered due to the high proximity of the sensor to the target. The actual temperature of the plate is not required as the correction is only of a
"geometric" type.

Figure 7 (upper panels) shows an example of BB and NB measurements collected at "Etna-Montagnola" (37.72° N, 15.00° E, 2600 m asl) on 30 August 2024 05:22 UTC, showing the degassing plume of Mt. Etna. The NB image (Fig. 7b) shows clearly the presence of the "ghost" effect approximately in the centre of the image.





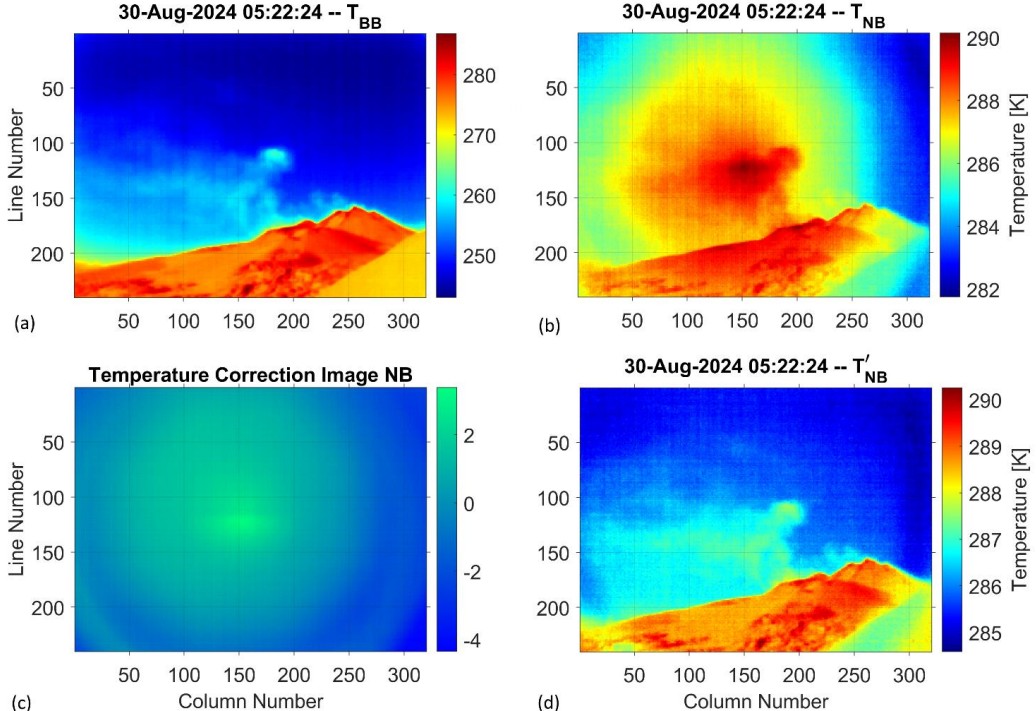

**Figure 7: TIR measurements collected at "Etna-Montagnola" on the morning of 30 August 2024. (a) BB and (b) NB original temperatures. (c) The temperature correction image that was obtained from black target measurements. (d) NB temperatures after the removal of the ghost image.**

A similar "ghost image" is present in the black target measurement too, so a "correction image" (Fig. 7c) is simply obtained as the difference between the black target temperature image $T_{b,NB}(i,j)$ and its mean value $\overline{T_{b,NB}}$ over the 320 x 240 array:

$$dtc_{NB}(i,j) = T_{b,NB}(i,j) - \overline{T_{b,NB}} \qquad (13)$$

To remove the "ghost" effect for any image we deduct the correction factor (Eq. 13) from the temperature originally measured in each pixel:

$$T'_{NB}(i,j) = T_{NB}(i,j) - dtc_{NB}(i,j) \qquad (14)$$

where $T_{NB}(i,j)$ is the original temperature measured and $T'_{NB}(i,j)$ the corrected temperature (Fig. 7d).

The blackbody measurements were usually performed at the start and about after 1 hour of scene acquisition. This simple geometric correction does not always give optimal results; in some cases, the "ghost" effect is only partially removed, and some noise effects remain in the scene images. This is probably due to the different temperatures of the black target and the collected scene, as well as to any variations in the internal temperature of the instrument. Generally, nighty measurements are less affected by these errors.





### 4.2 Matching MODTRAN with Broadband Clear Sky Temperatures

In principle, the BB temperatures don't have to be corrected, because of the factory calibration of the sensor (the digital counts to temperature conversion). Currently, uncooled microbolometer IR cameras with detection sensitivity between 8 µm and 14 µm are optimised for temperatures between 270 K and 370 K, which are higher than the average clear sky and most of the clouds' temperatures. The temperature accuracy guaranteed by the manufacturer is the greater of $\pm$ 5 °C or $\pm$ 5 % between 5 °C to 140 °C and $\pm$ 10 % between 140 °C to 330 °C (https://www.thermal.com/uploads/1/0/1/3/101388544/mosaic_core_specification_sheet_2021v2-web.pdf, last access: 7 January 2025). The calibration of the BB channel was already confirmed in the laboratory between 270 K and 300 K using a custom-built blackbody plate of high emissivity ($\epsilon > 0.98$) that could be heated and cooled using a thermoelectric (Peltier) cooler that operates by applying a current to a semiconductor material (Prata et al., 2024). Conversely, at lower temperatures, the calibration is not optimal and the temperature of the chalcogenide lens (Fig. 1b), placed in front of the sensor, becomes predominant. Moreover, the lower limit detectable for this type of sensor is 230-240 K, while the clear sky temperatures could easily reach lower values, especially at high altitudes, where generally the volcano measurements are carried out. The minimum temperature we ever measured was $T_{BB}$ = 230.6 K at Sabancaya volcano (Perù) at 5000 m asl.

To make the quantitative retrieval effective, we used MODTRAN 5.3 to simulate the temperature measured by the TIR cameras. Even if a proper atmospheric Pressure, Temperature, and Humidity (PTH) vertical profile is used (near in space and time to the VIRSO2 acquisitions), the scene collected by TIR cameras represents the real local atmospheric situation in a volcanic environment, with the presence of gases and particles (ash and water droplets) into an atmosphere which is not simple to simulate with a radiative transfer model (RTM). This, together with the low-temperature limits of BB sensor, causes very often the clear sky $T_{BB}$ to be higher than those simulated by the model. In Fig. 8a the green crosses represent the minimum temperatures computed row by row from $T_{BB}$ for the scene of Fig. 7a. To avoid problems related to pixels affected by the plume and too close to the land (with higher sky temperatures due to heating of the ground), only the lines from 1 to 150 were considered. The blue line represents the simulated MODTRAN clear sky temperatures, as a function of the elevation angle $\theta$ (Eq. 1). The atmospheric PTH vertical profile on 30 August 2024 at 6 UTC (37.5° N, 15.0° E) from the National Centers for Environmental Prediction (NCEP) dataset (Kalnay et al., 1996) was used in the model and the top of the atmosphere (at 100 km) was considered not emitting ($T_s$ = 0 K). The clear sky $T_{BB}$ are about 10-20 K higher than the simulated temperatures. To correct this gap, it is not possible to simply add or subtract a certain value (depending on $\theta$) to all the pixels of the image. The plume pixels are characterised by higher temperatures than those of clear sky; depending on their opacity, the difference with MODTRAN can be smaller or even absent.

It is possible to modify $T_{BB}$ below a certain temperature making some reasonable hypotheses about the factory calibration and the characteristics of the chalcogenide lens or, alternatively, bring closer MODTRAN simulations data to clear sky measurements, adding aerosol or water vapour to the simulated atmosphere, even though this is not always enough. A simple and valid method to match the MODTRAN simulation with the $T_{BB}$ image is instead to consider the top of the atmosphere as an emitting layer above the plume. This layer may represent either the sensor's inability to go below certain temperatures, and/or the effect of high cirrus clouds. The presence of the layer at temperature $T_s$ does not imply a generic additive term which could lead to erroneous results, such as that due to temperatures higher than those of the air for extremely opaque plumes. Indeed, an increase in the columnar contents (ash or $SO_2$) of the plume progressively reduces the effect of the presence of the $T_s$ layer. In this case, the global clear sky radiance $L_s$ as a function of $\theta$ that reaches the camera is:

$$L_s(\theta) = \varepsilon_s \cdot B(T_s) \cdot \tau_0(\theta) + L_0(\theta) \tag{15}$$





where $\varepsilon_s$ and $T_s$ are the emissivity and the temperature of the sky at the top of the atmosphere, $\tau_0$ and $L_0$ are the atmospheric clear sky transmittance and path radiance (obtained from MODTRAN). Assuming that there is at least one pixel inside the 320 x 240 array unaffected by the plume (clear sky), it is possible to compute the clear sky radiance $L_s$ by applying the Planck formula to the absolute minimum of $T_{BB}(i, j)$. Knowing the specific elevation angle of the minimum used, now the $T_s$ value, needed for matching the simulated and the measured BB temperatures, can be easily computed from Eq. 15, assuming for the sky $\varepsilon_s = 1$ (blackbody). Finally, the overall MODTRAN Look-Up Tables (all angles, all different plume columnar contents, both BB and NB channels, see the detailed description in Sect. 5) can be obtained from Eq. 15 by using the computed temperature $T_s$ and the specific overall transmittance and path radiance. In Fig. 8a the red line represents the MODTRAN clear sky temperatures computed with $T_s = 226$ K, matched to the minimum $T_{BB}$. Note the border effect of BB data that, in this case, produces an unrealistic increase (about 1-2 K) of the clear sky temperature for angles greater than 40°.

Although it involves a longer calculation time, it is advisable to carry out this procedure ($T_s$ estimation and LUTs recomputation) for every VIRSO2 image, to take into account possible changes to the local atmospheric situation and/or instrument's internal temperature variations. In Figure 8b the comparison between BB and NB MODTRAN clear sky simulations is shown, together with the corresponding VIRSO2 BB and NB measurements. The cyan crosses symbols represent the NB temperatures after the final re-calibration (see Sect. 4.3).

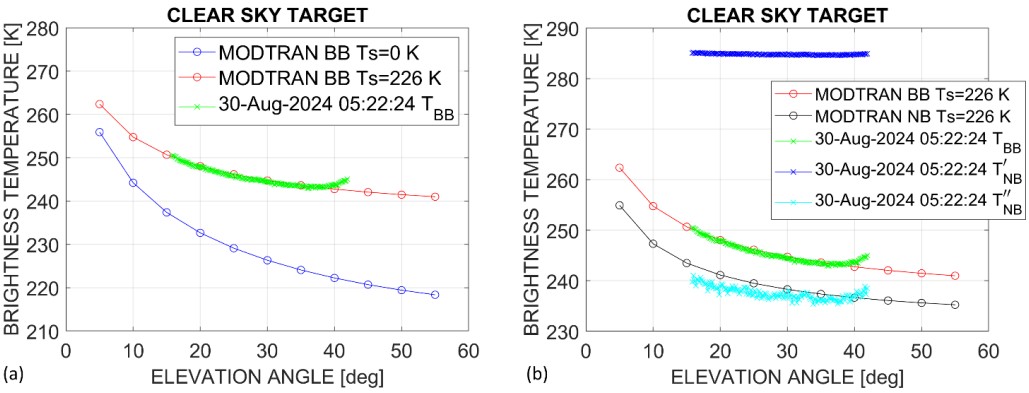

**Figure 8: Etna-Montagnola 30 August 2024 05:22 UTC clear sky test case. (a) Comparison between MODTRAN BB simulations with Ts = 0 K (blue line) and Ts = 225.8 K (red line) and VIRSO2 $T_{BB}$ (green crosses). (b) Comparison between MODTRAN BB (red line) and NB (black line) simulated temperatures with the corresponding VIRSO2 measurements. The blue and cyan crosses represent the NB temperatures before and after the calibration procedure respectively.**

### 4.3 NarrowBand calibration

As described in Sect. 4, the presence of the 8.7 μm filter placed in front of the camera strongly affects the measurements, destroying any previous factory calibration of the broadband sensor. In Fig. 7a, $T_{BB}$ min/max variation is more than 40 K, ranging from about 242 K (sky) to 287 K (land), while $T'_{NB}$ (the NB temperature after the "ghost" effect removal, Fig. 7d) ranges only from about 285 K (sky) and 290 K (land). The very limited $T'_{NB}$ variation (only about 5 K) makes the calibration particularly important and critical at the same time. The need for this second correction is also evident from the comparison between clear sky $T'_{NB}$ and MODTRAN (blue crosses and black line in Fig. 8b).

In principle, the NB measurements can be written as:





$$B(T'_{NB}) = \tau_f \cdot B(T''_{NB}) + (1 - \tau_f) \cdot B(T_f) \tag{16}$$

where $\tau_f$ and $T_f$ are the transmittance and the temperature of the 8.7 μm filter and $T''_{NB}$ is the temperature of the target, i.e. the expected re-calibrated NB temperature. While $\tau_f$ is approximately known (about 0.8), the $T_f$ is unknown and variable: the filter is outside of the VIRSO2 system and therefore exposed to air, wind, and sun, but at the same time is near to the camera, so it is affected by internal temperature variations too. Moreover, to use it for re-calibration, $T_f$ must be known with an extremely good precision, given that usually it gives the greater contribution to Eq. 16, being $T_f \gg T''_{NB}$. For this reason, we decided to use a similar, but easier, numerical re-calibration method based, in first approximation, on a simple linear relationship:

$$T''_{NB}(i,j) = a \cdot T'_{NB}(i,j) + b \tag{17}$$

Given that there are two coefficients to be derived ($a$ and $b$), a minimum of two different calibrated temperatures $T''_{NB}$ are needed, preferably covering a wide range of temperatures, including those of the volcanic plume. A good choice for the lower one is the minimum sky temperature already used in the previous Sect. 4.2 (yellow box in Fig. 9a). For this point, we already know the expected value for the NB channel from the MODTRAN LUTs recomputed with $T_s$ = 226 K. Generally, the NB clear sky temperature is lower than the BB temperature of about 5-10 K, depending on the atmospheric composition, the ground altitude, and the elevation angle of the camera.

On the other hand, a good candidate for the higher temperature value can be the average of a small ground area (white box in Fig. 9a). Land temperatures are indeed usually higher than those of the plume and, by choosing the area far enough from the crater, it will not be affected by $SO_2$. Specific MODTRAN simulations computed considering a "grey" target, allow us to obtain the theoretical relationship between the BB and NB ground temperatures, which depends mainly on the ground temperature and the distance of the volcano from the camera, as well as the atmospheric composition (Fig. 9b).

Assuming the $T_{BB}$ as correct, the calibrated $T''_{NB}$ for sky and ground are:

$$T''_{NB}(sky) = T_{BB}(sky) - \delta T_S \tag{18}$$
$$T''_{NB}(ground) = T_{BB}(ground) - \delta T_G \tag{19}$$

where $\delta T_S$ and $\delta T_G$ are the sky and the ground temperature values to be subtracted from the BB to obtain the corrected NB temperature. As already reported in Sect. 4, we compare BB and NB pixels in the same scene position where the NB array must have been previously co-registered to BB. In the specific test case presented (Etna-Montagnola, 30 August 2024), with the VIRSO2 camera placed at 2600 m asl and at 3.5 km from the crater, $\delta T_S$ = 6.2 K and $\delta T_G$ = -0.1 K, from which the linear calibration coefficients of eq. 17 become: $a$ = 9.76 and $b$ = −2542. Fig. 9a shows the resulting calibrated NB image, now ranging from 236 K to 290 K.



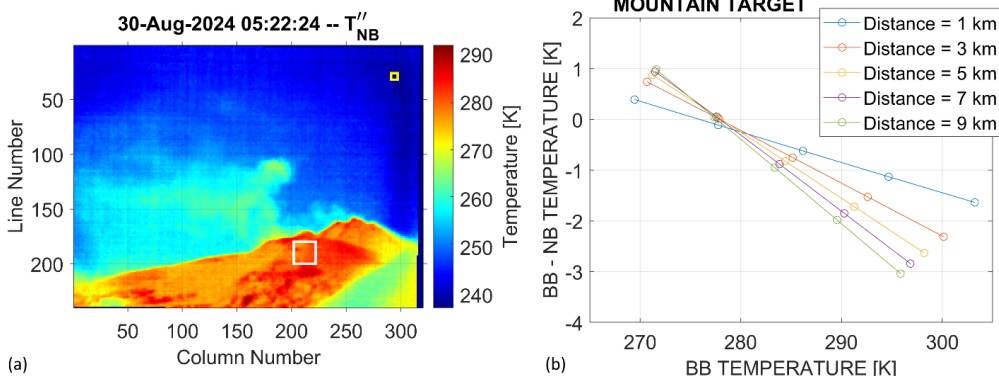

**Figure 9: Etna-Montagnola 30 August 2024 05:22 UTC test case. (a) Calibrated and co-registered $T''_{NB}$ temperature (the white and yellow box indicate the pixels considered in the ground-sky calibration). (b) Ground simulated $\delta T_G$ values for a target placed at different distances (1-9 km) from the camera ($\theta = 10°$). The target has a uniform emissivity ($\varepsilon_G = 0.95$) and a temperature range of 270-310 K**

### 4.4 "Zero image" and "Temperature Difference" computation

Once the re-calibration and the other correction procedures have been applied, a temperature difference image is made, consisting of the difference between the corrected image (both BB and NB) and a "zero value" corresponding to imagery with no plume. These images (Fig. 10), labelled $DT_{BB}$ and $DT_{NB}$, are the main information content of the retrieval scheme, because they will be used to obtain the plume columnar contents by comparing them with the simulated MODTRAN values.

$$DT_{BB}(i,j) = T_{BB}(i,j) - T_{0,BB}(i) \tag{20}$$

$$DT_{NB}(i,j) = T''_{NB}(i,j) - T''_{0,NB}(i) \tag{21}$$

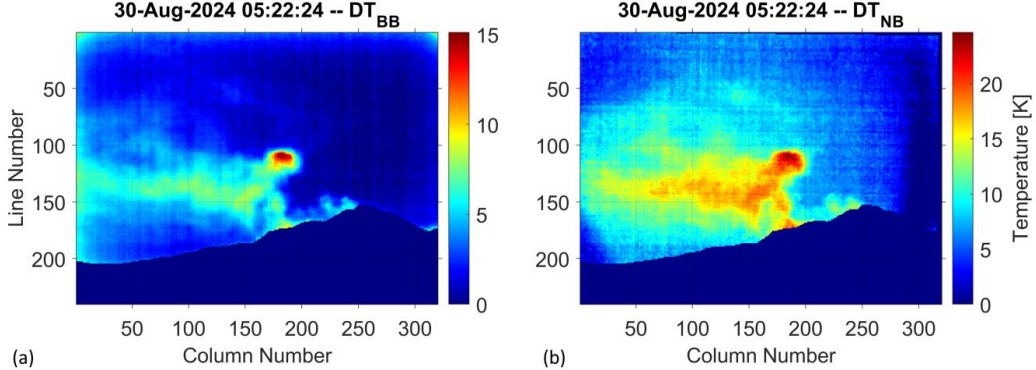

**Figure 10: (a) BB and (b) NB temperature difference images (Etna-Montagnola 30 August 2024 05:22 UTC).**

Obtaining the zero images is not so easy in practice as they should contain no plume and yet image the area of the sky close to where there is a plume so that a similar part of the atmosphere is sampled. Moreover, because also the clear air temperatures are not uniform, and increase as the $\theta$ angle decreases (see Fig. 8), it is necessary to consider a "zero value" for each row of the image. It is possible to simply consider the smoothed row by row minimum values of the overall




image or the row by row average values of a specific plume-free sub-region of the image. Alternatively, it is also possible
to use the clear sky temperatures simulated with MODTRAN after the "matching" procedure (red and black lines of Fig.
8b). These tend to be less noisy, but on the other hand they can produce some small biases in the DT values if the matching
VIRSO2-MODTRAN is not accurate. The retrievals are very sensitive to the variations of DT, so an imprecise "zero
image" computation can cause large retrieval errors.

**5 SO₂ Retrieval**

Using MODTRAN, we can simulate the behaviour of BB and NB temperatures according to different SO$_2$ vertical
columnar densities and elevation angles. This is done by inserting in the atmospheric vertical profile a horizontal layer
with increasing SO$_2$ vertical column density (VCD) contents. This configuration ("MODTRAN plume" in Fig. 11) would
be correct only if the plume moved exactly in the direction of the camera ($\omega = 90°$). If instead the wind direction is

parallel to the camera's focal plane ($\omega = 0°$), to obtain a correct configuration of the simulation, it is necessary to make
a conversion from the "real plume" to the "MODTRAN plume" geometry (from vertical to horizontal). In Fig. 11 a
scheme of the problem is shown. As $\theta$ increases, it is necessary to increase the height and the thickness of the MODTRAN
layer where the plume is present, to make sure the path length of the radiance is correct (see the segments CP$_1$, CP$_2$, P$_1$Q$_1$,
and P$_2$Q$_2$ in Fig. 11 which are the same for both the configurations).

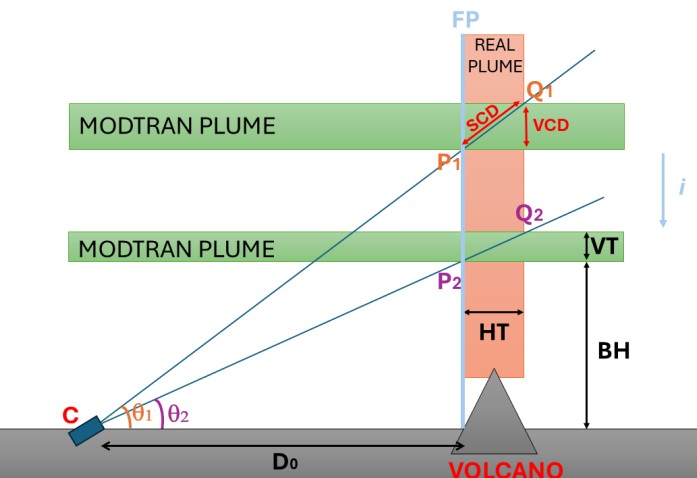

**Figure 11: Example scheme of the real vs MODTRAN plume configuration if the wind direction is parallel to the camera's focal plane. The VIRSO2 camera is in C and the focal plane (FP) is perpendicular to the figure.**

These simple equations were used to compute the bottom and top altitude of the MODTRAN plume layer:

$$BH = \left(D_0 - \frac{HT}{2}\right) \cdot tan(\theta) + h_0 \tag{22}$$

$$VT = HT \cdot tan(\theta) \tag{23}$$

where $h_0$ is ground altitude, $BH$ is the bottom height of the plume (asl), $HT$ and $VT$ are the horizontal and vertical thickness
of the plume, $D_0$ is the horizontal distance to the target. If the wind direction is not parallel to the focal plane, that is $\omega \neq$
$0°$, as first approximation $D_0$ can be simply replaced by the average $D_0^*$ value (see Sect. 3.1). $HT$ is an input parameter
which is usually unknown, but it does not significantly affect the results (see Sect. 7).



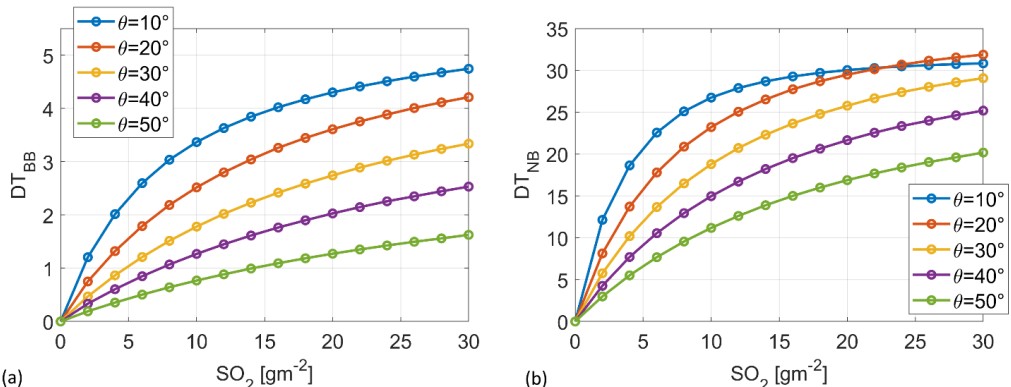

**Figure 12: BB (a) and NB (b) MODTRAN *DT* vs SO2 VCD performed for the Etna-Montagnola testcase (30 August 2024).**

Figure 12 shows the simulated $DT_{BB}$ and $DT_{NB}$ vs SO$_2$ *VCD*, varying the viewing angles $\theta$ from 10° to 50°, for the Etna-Montagnola test case (30 August 2024 06 UTC NCEP PTH profile, $h_0 = 2600$ m, $D_0^* = 3$ km and $HT = 200$ m) and MODTRAN LUTs computed with $T_s = 226$ K. It is clear a strong dependence of *DT* on elevation angles. In case of presence of SO$_2$ only (no particles in the plume), in the BB image the plume should be poorly visible as $DT_{BB}$ ranges only between 0 K and 5 K. This is not the case of Fig. 10a, clear sign that in the depicted emission there are also some particles (ash and/or water droplets). The NB channel is instead much more sensitive to SO$_2$, with $DT_{NB}$ that ranges between 0 K and 32 K. Knowing $\theta(i,j)$ for each pixel of the image, by means of a simple bilinear interpolation the final image of SO$_2$ column content can then be computed. Figure 13a shows the SO$_2$ *VCD* obtained as described, that is attributing all the NB signal ($DT_{NB}$) to SO$_2$ only. Note that to mitigate the inaccurate "ghost" image removal (Fig. 10b), only the pixels with $DT_{BB} > 2$ K were considered.

Here the presence of particles in the plume (ash and/or water droplets) was not considered, while they strongly affect both BB and NB channels. Fig. 13b shows a close in time picture of the volcanic plume in which the presence of condensed water vapor particles seems highly probable. The simplification to attribute all the NB signal ($DT_{NB}$) to SO$_2$ only, produces an overestimation of the results obtained. Removing the contribution of the particles from $DT_{NB}$ will be an important improvement of the algorithm in the future.

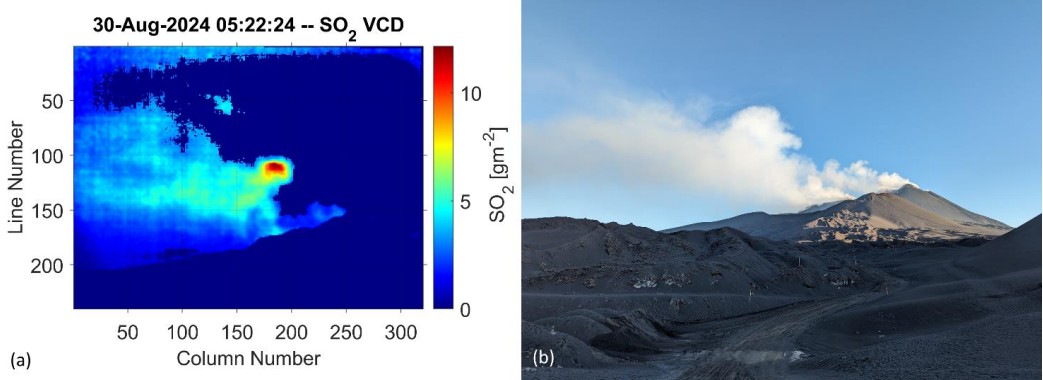

**Figure 13: Etna Montagnola 30 August 2024 (a) 05:22 UTC SO$_2$ VCD image (g/m²) obtained from NB channel, considering SO$_2$ only (not corrected for the particles). (b) A picture of the volcanic plume at 05:00 UTC**




**6 SO₂ Mass and Flux Computation**

The comparison with MODTRAN gives, for each pixel of the image, the SO₂ *VCD* content. The computation of SO₂ mass ($M_s$) and flux ($F_s$) requires converting the "MODTRAN" vertical content to the "real" slant column density (*SCD*, Eq. 24 and Fig. 11). Now the SO₂ total emitted mass can be simply computed multiplying the *SCD* for the area of the pixels (Eq. 25, Tamburello et al., 2012).

$$SCD = VCD/sin(\theta) \tag{24}$$
$$M_s = \sum_{i,j} a(i,j) \cdot SCD(i,j) \tag{25}$$

In this way, it is possible for example to compare VIRSO2 results with other ground-based sensors such as UV cameras, paying attention to consider the same FOV area. Instead, to validate the VIRSO2 results with space-based data (satellite), because of the very different point of observation and spatial/temporal resolution, the SO₂ flux is a better quantity to be used. Considering a transect perpendicular to the direction of the plume (vertical, horizontal, or oblique), the SO₂ flux (function of time) is computed:

$$F_s(t) = \sum_{i=1}^{N} v_i(t) \cdot SCD_i(t) \cdot l_i \tag{26}$$

where $i$ (from 1 to $N$) are the pixels contained in the transect, $v$ is the plume velocity and $l$ is the pixel width (orthogonal to the flux direction). The plume velocity can be computed pixel by pixel and time by time using an optical flow algorithm or simply obtained from the focal plane component of the wind speed (*ws*) and considered the same for all the plume pixels (Eq. 27)

$$v = ws \cdot cos(\omega) \tag{27}$$

In this case (Etna-Montagnola 30 August 2024) the volcanic emission is quite low, and the optical flow doesn't give reliable results. So, the wind speed was set to 2.1 m s⁻¹, obtained as the average speed at 3.5 km asl at Etna (37.5° N; 15.0° E) on 30 August 2024 5:00-7:00 UTC from ERA5 hourly dataset. In the same way, the average wind direction was set to 38 degrees north, so $\omega \approx 30°$ and the plume velocity becomes $v = 1.8$ m s⁻¹. Finally, it is recommended to choose a transect not too close to the crater because in the RTM simulations the plume is assumed to be in thermal equilibrium with the surrounding air temperature. For this reason, the VIRSO2 flux is computed as the average of the fluxes obtained considering the 75th, 100th and 125th columns as vertical transects. The first 40 lines were discarded due to some noise/residual error (ghost effect) and because the plume stayed generally in the lower part of the images for this test case (see Fig. 13a and BB temperature videos in Supplementary Materials).

The TROPOspheric Monitoring Instrument (TROPOMI) is a spectrometer on board Sentinel-5 Precursor (S5P) polar orbit satellite that covers a spectral range from ultraviolet (UV) to short wave infrared (SWIR), with a spatial resolution at the nadir of 5.5 x 3.5 km² and a revisit time of about 1 day (Veefkind et al., 2012). Particularly significant for TROPOMI UV bands is the ability to detect SO₂ columnar abundance of about 0.02 g m⁻² (0.7 DU), i.e., about thirty times better than the sensitivity of the multispectral satellite sensors (Corradini et al., 2021). Here the near real time (NRT) Level 2 SO₂ product collected on 30 August 2024 at 11:33 UTC is considered. The SO₂ *VCD* image (Fig. 14a) is obtained by interpolating at 3.5 km asl the two "sulfur dioxide total vertical column" products at 0.5 km agl and 7 km asl. Finally, the SO₂ flux is computed by applying the traverse method, and the wind speed is the same considered for the VIRSO2 (*ws* = 2.1 m s⁻¹). Different from the camera, all the transects perpendicular to the volcanic cloud axis are considered (Pugnaghi



et al., 2006; Merucci et al., 2011; Theys et al., 2013; Corradini et al., 2021). Depending on wind speed and plume extension, this makes it possible to obtain an SO$_2$ flux trend for several hours before the image acquisition (shown partially in Fig. 14b).

The SO$_2$ fluxes obtained from the TIR camera and TROPOMI are comparable, with the first one greater than about 300-800 t d$^{-1}$ (tons per day, Fig. 14b). The mean values are 481 t d$^{-1}$ and 996 t d$^{-1}$ for TROPOMI and VIRSO2 respectively. The VIRSO2 retrieval clearly shows the fluctuations in emission, which are not visible from satellite. Considering a ±40 % error for both VIRSO2 (see Sect. 7) and TROPOMI (±35 % retrieval error (Corradini et al., 2021) plus ±20 % for wind speed, quadrature sum), the data mainly fall within the uncertainty of the procedure. In this case, as said in Sect. 5, the overestimation of VIRSO2 is probably due to the presence of water vapor particles condensed in the volcanic plume.

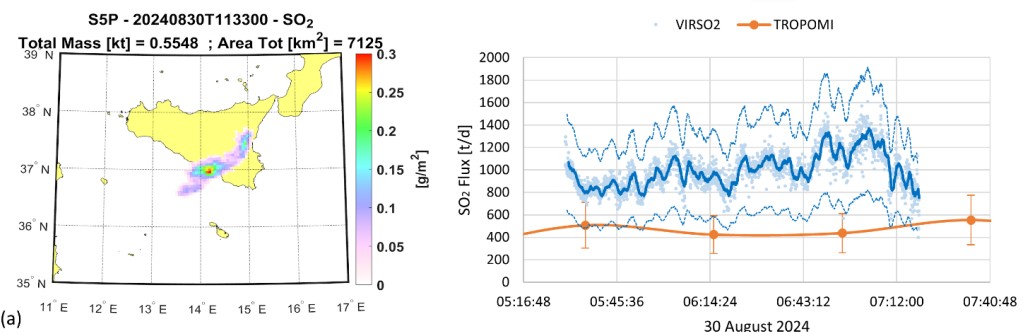

**Figure 14: Etna Montagnola 30 August 2024. (a) The S5P-TROPOMI SO$_2$ VCDs map (11:33 UTC) from which the SO$_2$ flux has been computed. (b) SO$_2$ Flux (in tons per day) obtained from TROPOMI (orange line and symbols) and from VIRSO2 (cyan symbols and blue lines). The blue solid line is a moving average over 1 minute, while the shaded ones are obtained considering an error of ± 40 % on the retrieval.**

## 7 Sensitivity Study

Due to the various steps and inputs of the procedure, there are several possible sources of error, not all of which are easily quantifiable. For example, the ghost image removal, the NB channel recalibration, and the "zero" image computation can introduce errors that cannot be assessed a priori and that can have an impact on the SO$_2$ VCD retrieval. Other sources of errors, related to the uncertainties of the input parameters, are more easily computable.

The sensitivity study was done by evaluating the total mass erupted on 30 August 2024 between 05:30 UTC and 07:18 UTC and obtained by integration of the SO$_2$ flux. For the test case considered, the reference configuration, its variability and the related errors are shown in Table 2. The impact on SO$_2$ of any errors in the MODTRAN simulations was evaluated through the first three input parameters ($HT$, $D_0$ and $PTH$). As Table 2 shows, while the geometrical configuration ($HT$ and $D_0$) uncertainties do not lead to significant retrieval errors, the use of an appropriate atmospheric vertical profile ($PTH$) is important. In this case, the August and September 1981-2010 monthly mean profiles around Etna (32.5° N - 42.5° N; 10.0° E - 20.0° E) were considered instead of the specific date (30 August 2024, 06 UTC), obtaining an error of -7.6 % and 14.2 % respectively. The central elevation angle ($\alpha$) and the wind direction affects mainly the size of the pixels of the image, which is important for the computation of both SO$_2$ total mass and flux (Eq. 25-26). $\alpha$ is a very sensitive parameter because it affects the $SCD$ computation too (Eq. 24), but it can be obtained usually with good precision using an inclinometer attached to the VIRSO2 camera or from the altitude of a known reference point in the image, for example the top of the volcano (Snee et al. 2023). The same accuracy cannot be achieved for the wind direction. This latter parameter can be obtained from reanalysis data (as in this case) or directly from in-situ measurements, but the local effects




related to orography can produce significant variations. Finally, the plume speed is directly proportional to the flux (see Eq. 26), so its uncertainty proportionally affects the flux.

In summary, an overall uncertainty of about ±40 % can be calculated as the sum in quadrature of the maximum (in absolute value) individual uncertainties.

| Input Parameter | Reference Value | Range of variability | SO₂ Flux Error |
|---|---|---|---|
| Horizontal thickness of the plume ($HT$) | 200 m | 100 - 1000 m | -0.4 % / +0.2 % |
| Horizontal distance to the target ($D_0$) | 3 km | ± 500 m | ± 4 % |
| Atmospheric conditions ($PTH$) | NCEP 30 Aug 2024 06 UTC | NCEP Monthly Mean 1981- 2010 August & September | -8 % / +14 % |
| Central Elevation Angle ($\alpha$) | 21° | ± 2° | -26 % / +12 % |
| Wind Direction | 38° north | ± 10° | -13 % / +12 % |
| Plume Speed ($v$) | 1.8 m s⁻¹ | ± 20 % | ± 20 % |

Table 2: Etna-Montagnola 30 August 2024 (2600 m asl). SO2 flux errors related to the input parameters of the procedure.

**8 Conclusions and Future Work**

In this paper, we present a new TIR ground-based system for the detection of volcanic plumes and the retrieval of geometry and SO₂ emissions. The main advantages of this instrument are the high portability, the relatively low cost, and the possibility to make measurements during both day and night. The drawbacks are mainly related to the calibration: the presence of the filter centred at 8.7 μm outside of the system produces some critical effects that need to be corrected through a post-processing procedure. The overall retrieval procedure consists of three different steps: first the instrument

view geometry is computed, then the calibration is carried out and finally the SO₂ is retrieved. Every step of the process introduces errors: here an overall error of ±40 % was estimated for the SO₂ flux retrievals by considering the uncertainties on the instrument viewing geometry and the atmospheric profiles of pressure, temperature, and humidity.

As test cases, some VIRSO2 measurements were performed at Mt. Etna in March-April 2021 and August 2024. The measurements collected in 2021 show the effectiveness of the plume detection and geometry retrievals and the importance

of the wind correction, while the 2024 SO₂ flux retrievals show comparable results with those obtained from space using the S5P-TROPOMI UV sensor despite the huge difference between the two instruments (spectral range, spatial resolution, point of view). In particular, the higher values of SO₂ flux obtained from the camera compared with TROPOMI, could be due to the presence of particles (ash or water droplets) in the volcanic plume that cause an overestimation of the VIRSO2 retrieval. The aim of the article is mainly methodological, that is to explain in detail the algorithm developed and used for

the SO₂ estimation. The comparison and validation with other instruments, both satellite and ground-based, will be the subject of other papers (in preparation) in which the data collected at Etna, Stromboli, Sabancaya (Peru), and Popocatepetl (Mexico) volcanoes will be used.

Several improvements can be included in the procedure in the future. A low-temperature blackbody instrument, composed of a uniform emissive head, a refrigerating unit, and an electronic temperature controller, would be very useful to

535 understand the behaviour of the BB sensor below 270 K, which is outside of the factory calibration range of the camera. The ghost image removal procedure presented in this paper is very simple but sometimes not completely effective due to the temperature difference between the black target and the real detected scene. The incomplete elimination of the "ghost" effects can produce errors in the following recalibration of the NB 8.7 μm channel which is the crucial part of the algorithm, so an improvement of this part would be desirable. The addition of temperature sensors near the filter and

540 inside the camera body would help and be quite easy to implement. Finally, the presence of particles in the volcanic plume



(ash, water droplets, ice, etc …) strongly affects the NB 8.7 μm measurements. The correction for particles' contribution will represent an important improvement of the algorithm.

The $SO_2$ flux is a fundamental parameter for the study of the volcano's activity and for the comparison with other instruments, both from ground and space. To compute it, the velocity of the plume is a necessary value to be known; in this sense, an optical flow algorithm could give important information regarding the different speeds of the various parts of the plume, both in time and space.

**Author contribution**

LG and SC Writing – original draft preparation; LG and FP Methodology; LG Software and MODTRAN simulation; LG Visualization; LG, SC, DS and LL Investigation (data collection); RB and LM resources, supervision, funding acquisition; All Writing – review & editing.

Competing interests. The authors declare that they have no conflict of interest.

**Acknowledgments**

The authors gratefully acknowledge Sergio Pugnaghi (retired researcher of Modena and Reggio Emilia University, Italy) for his valuable support in the development of the procedure for the retrieval of $SO_2$ and Paul A. Jarvis (p.jarvis@gns.cri.nz) for the help provided in the comparison of our plume height results with those of the procedure described in Snee et al. 2023, used as validation in Sect. 3.1. Finally, Dr Cirilo Bernardo (AIRES Pty Ltd) is thanked for help in designing and building VIRSO2.

**Financial support**

- INGV-MUR Project Pianeta Dinamico - DYNAMO project (DYNAmics of eruptive phenoMena at basaltic vOlcanoes).
- VESUVIO project (Volcanic clouds dEtection and monitoring for Studying the erUption impact on climate and aVIatiOn) funded by the Supporting Talent in ReSearch (STARS) grant at Università degli Studi di Padova, Italy.

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
