# Peer review of "A Novel Simplified Ground-Based TIR System for Volcanic Plume"

_EGUsphere, 2025_

## Author Comment (AC1)

**Reviewer Comments**

**eguphere-2025-63**

**Guerrieri et al., A Novel Simplified Ground-Based TIR System for Volcanic Plume Geometry, SO₂ Columnar Abundance, and Flux Retrievals**

We would like to sincerely thank the reviewer for his work and the time spent reading our article.

His/her suggestions have helped to significantly clarify some parts of the paper.

The line numbers in the authors' comments refer to the new manuscript revised according to the reviewers' suggestions.

This manuscript is comprehensive and well-written. My concerns are focused on the theoretical basis of the calibration and retrieval routines, as discussed below. I recommend that the manuscript be revised to address these concerns.

I am not familiar with the transmission spectra of chalcogenide glasses, but I do know that chalcogenide glasses contain sulfide compounds. Are the authors certain that the glass transmission spectra do not contain features that overlap with the SO₂ spectra?

Yes, the reviewer is right. Chalcogenide glasses are widely used in infrared spectroscopy due to their wide window of transparency but they can contain sulfur, selenium and tellurium (but not oxygen). We don't know exactly the real composition of our glass so we don't know its transmission spectra. As written at Line 78: *"a simple top-hat function between 1282 cm-1 (7.8 µm) and 714 cm-1 (14 µm) was considered for BB since there is no specific information about it from SEEK Thermal".* Some transmission functions of Chalcogenide glasses can be found in literature (see for example Calvez, 2017 or https://www.hypoptics.com/components/infrared-lenses/chalcogenide-lenses/) and there are no features at 8-10 microns. In fact if gas bubbles were trapped in the glass (it has to be SO2 gas) then you would expect H2O too.  Then you would expect chemical conversion to acid and the glass would deteriorate. The reason that glass is used is because none of those things happen.  But let's suppose there are absorptions from something (it can't be SO2), but it could be something that lowers the transmission. But this effect is permanent, that is the transmission function doesn't change with time or with whatever the camera is looking at. In the BB this effect should have already been considered in the calibration performed by the manufacturer. In the NB this effect is completely eliminated by the recalibration of temperatures performed by our procedure (Sect. 4.3).

Laurent Calvez. Chalcogenide glasses and glass-ceramics: Transparent materials in the infrared for dual applications. Comptes Rendus. Physique, 2016 Prizes of the French Academy of Sciences /Prix 2016 de l'Académie des sciences, Volume 18 (2017) no. 5-6, pp. 314-322. doi : 10.1016/j.crhy.2017.05.003. https://comptes-rendus.academie-sciences.fr/physique/articles/10.1016/j.crhy.2017.05.003/

Conceptually, the VIRSO2 camera detects SO₂ absorption as differences between the narrow band (NB) and broad band (BB) images of plumes.

We don't detect SO2 by making a difference between narrow band (NB) and broad band (BB) images. In our procedure, the SO2 content is obtained only from the narrowband (NB) image, by comparing the $DT_{NB}$ image with the MODTRAN simulated values. This misunderstanding can be due to an inaccurate explanation of this part of the procedure. For this reason, we added some sentences in Sect. 4.4 and Sect 5 to clarify this aspect:

Line 381:
*Once the re-calibration and the other correction procedures have been applied, a temperature difference image is made, consisting of the difference between the NB corrected image and a "zero value" corresponding to imagery with no plume (Eq. 20):* $DT_{NB}(i,j) = T''_{NB}(i,j) - T''_{0,NB}(i)$
Line 391:
*These images (Fig. 10), labelled $DT_{BB}$ and $DT_{NB}$, are the main information content of the retrieval scheme; in particular, the $DT_{NB}$ image will be used to obtain the SO2 plume columnar contents by comparing it with the simulated MODTRAN values.*

Line 453:
*Knowing $\theta(i,j)$ for each pixel of the image, by means of a simple bilinear interpolation the final image of SO2 column content can then be computed from $DT_{NB}$ image.*

Implicit in this approach are the assumptions that (1) the BB measurements are not affected by the presence of SO₂, and (2) there are no differences between BB and NB measurements for a clear-sky optical path. As to the first assumption, we see from Fig. 1c that the BB spectral response covers the SO₂ absorption feature at 8.7 μm. The plumes are evident in every image of BB temperatures presented by the authors, indicating that the presence of SO₂ had increased the optical depth of the plumes, relative to clear-sky paths. I do recognize that the presence of steam, ash, and other particulates will also increase the optical depth of a plume, but the authors are focused on SO₂.

The impact of the SO₂ absorption on BB radiance depends on gas concentration and the scene temperature, as the peak radiance can be coincident with the maximum gas absorption. SO₂ is transparent at wavelengths > 9.5 μm, and an alternate design for the BB camera would include a filter to block radiance at wavelengths < 9.5 μm. Admittedly, this design option would introduce a new filter to the calibration procedure.

The two assumptions said by the reviewer are not implicit in our procedure.

The theoretical dependence of SO₂ on BB temperatures is clearly shown in Fig. 12a, and this is extremely low compared to that on NB temperatures. In any case, as said above, the BB is not

used for the SO$_2$ content computation (only the DT$_{NB}$ image is used), so this is not a problem. The BB image is used only for the calibration of NB, using clear sky and ground temperatures from pixels not affected by the volcanic plume (Sect. 4.3).

Regarding the second assumption, the MODTRAN simulations of BB and NB radiance at 226 K shown in Fig. 8b demonstrate that the BB and NB do not agree for clear-sky paths. The authors attribute this difference to the transmission (and temperature) of the NB filter, but (presumably) the MODTRAN simulations were based on the normalized spectral response (Fig. 1c) rather than the transmission of 0.80 cited in the text. An alternate explanation for the differences between the NB and BB radiance is that the NB spectral response excluded the peak radiance for a sky temperature of 226 K whereas the BB spectral response included the peak radiance.

We don't assume that there are no differences between BB and NB measurements for a clear-sky optical path. As said by the reviewer, the theoretical differences between BB and NB temperatures (obtained from MODTRAN) are clearly shown in the fig. 8b (red and black lines). And these differences are considered and used for the NB calibration ($\delta T_S$ parameter in Eq. 18, Sect. 4.3). Clearly these theoretical differences are due to the BB and NB spectral response functions used in MODTRAN. We don't attribute this theoretical difference to the transmission and temperature of the NB filter, because they are not used in the MODTRAN simulation (the temperature of the filter is unknown). On the contrary, we attribute to the transmission and temperature of the NB filter the difference between the NB measurement and the NB theoretical (MODTRAN) value.

Fig. 8b shows that the corrected NB brightness temperatures (T'$_{NB}$) are ~45 K warmer than the MODTRAN NB simulations. This large disagreement raises concerns about the ghost image correction. The current approach is to calculate a correction factor based on the mean value of the black target image (Lines 230-235), but a correction factor based on the background values of the black target image (i.e., outside of ghost image) might be more appropriate.

In the figure below you can see a black target image obtained on 30 August 2024 at 05:18 UTC:

[Figure]

Black Target Image

The mean value of this image is 290.8 K, while the minimum value (outside the ghost image) is 286.5 K.

If we use the minimum value of the black target image instead of the mean value, we clearly obtain a slightly different temperature correction image $dtc_{NB}$ (see fig below, to be compared to fig. 7c in the manuscript).

[Figure]

dtc image, using the minimum value of the Black Target Image (range ~0 – 7 K)

Fig. 7c used in the manuscript, using the mean value of the Black Target Image (range ~-4 – 3 K)

If we used it to obtain T'$_{NB}$, we have that the disagreement between the NB measurement and

the NB MODTRAN clear sky simulation is about 40 K (see fig below) instead of about 45 K using the mean value of the black target image.

[Figure]

Using the minimum value of the Black Target Image

Fig. 8b used in the manuscript, using the mean value of the Black Target Image

So, the major contribution to the disagreement between the NB measurement and the NB MODTRAN clear sky simulation is not due to the ghost image removal method (as said by the reviewer) but to the temperature (and transmission) of the NB filter.

But the most important thing is that, as written at Line 231, the "ghost image" removal is only of a "geometric" type, so at this step the NB temperatures that we obtain are not important. The re-calibration of NB temperatures is performed later (Sect. 4.3). Both using the mean black target or the minimum black target, the final result ($SO_2$ columnar content) does not change (see fig below, obtained with the minimum black target value, to be compared to fig. 13a in the manuscript) because the simple linear equation used for the NB calibration (Eq. 17) changes accordingly (in this case for example the "b" parameter becomes b=-2498 instead of b=-2542).

[Figure]

30-Aug-2024 05:22:24 -- SO$_2$ VCD

Using the minimum value of the Black Target Image

Fig. 13a used in the manuscript, using the mean value of the Black Target Image

To clarify this important aspect, we inserted a new comment in the manuscript:

Line 253:
*"It's important to remark that at this step it doesn't matter if the resulting image $T'_{NB}$ is correct in temperature, because the temperature calibration of the NB is performed later (Sect. 4.3); so, for example, using the minimum value of the black target image instead of the mean one, it wouldn't change the final result, that is the SO2 content."*

Additional comments about the determination of sky temperature ($T_s$), based on Equation 15. To calculate the atm transmission ($\tau_o$) and, in particular, path radiance ($L_o$) MODTRAN needs a profile of atm temperatures. As discussed in Lines 271-274, NCEP profiles were used, with the top of the atmosphere (TOA) temperature set to 0 K. How, then, can $\tau_o$ and $L_o$ be used to determine $T_s$ with the TOA temperature of 0 K "baked in" to the MODTRAN simulations?

MODTRAN gives in output not only the total radiance that reaches the ground but also the singles components (atm transmission and path radiance). In particular, $\tau_o$ and $L_o$ depend only on the atmospheric vertical profile (PTH) used, not on the TOA temperature. For example, using the 30 August 2024 06 UTC NCEP PTH profile, a ground altitude of 2600 m asl, an inclination of 30 degrees, we obtain from MODTRAN and for the BB camera that:

$\tau_o$ = 0.634677

$L_o$ = 20.9123 mW m-2 sr-1 cm-1

both using Ts = 0K or Ts = 226K.

The total radiance that reaches the ground is instead Ls = 20.9123 mW m-2 sr-1 cm-1 for Ts = 0 K (it's equal to the path radiance obviously, according to Eq. 15) or Ls = 35.1779 mW m-2 sr-1 cm-1 for Ts = 226 K.

The authors make the enigmatic statement that the NB clear sky temperatures ($T_{NB}$) are 5- 10 K lower than the BB temperatures (Line 337), but do not show the $T_{NB}$ temperatures in Figs. 8a or b. I can infer that the comparison refers to the $T_{NB}''$ temperatures, which have been "corrected" for the ghost image and temperature/transmission of the 8.7 µm filter, but a reader shouldn't have to make such inferences. If the authors are referring to the $T_{NB}$ temperatures, then they should show these temperatures explicitly.

We are referring to the differences between the theoretical clear sky BB and NB temperatures obtained from MODTRAN and shown in fig. 8b (the red and black lines). We clarified this aspect in the manuscript:

Line 353:
*"Generally, the simulated NB clear sky temperature is lower than the BB temperature of about 5-10 K, depending on the atmospheric composition, the ground altitude, and the elevation angle of the camera (see the red and black lines in Fig. 8b)."*

[Figure]

Fig. 8b used in the manuscript

The BB Temperature Difference image ($DT_{BB}$) shown in Fig.10a provides further example that the BB measurements are sensitive to the presence of $SO_2$. The authors state that $DT_{BB}$ and the corresponding NB Difference image ($DT_{NB}$) are the foundations of the $SO_2$ retrieval scheme (Line 365) but do not explain why both inputs are necessary. MODTRAN simulations of both BB and NB are performed (Lines 386-387), but to what end? Are there two independent sets of $SO_2$ estimates, based on $DT_{BB}$ and $DT_{NB}$, that are compared to make a "composite" estimate? The sky and ground references temperatures for $T_{NB}$ are based on $T_{BB}$ (Eq. 18 and 19), but this formulation does not explain the use of both $DT_{BB}$ and $DT_{NB}$ in the retrieval procedure.

As said at the beginning of the reviewer's comments, this is a misunderstanding: for the SO2

content computation, only the $DT_{NB}$ and the corresponding NB MODTRAN simulations are used. As said above, we added some sentences in Sect. 4.4 and Sect 5 to clarify this important aspect.

Fig. 10a ($DT_{BB}$) was inserted in the manuscript to justify the presence of water or ash particles in the plume: as written at line 447, *"In case of presence of SO2 only (no particles in the plume), in the BB image the plume should be poorly visible as simulated $DT_{BB}$ ranges only between 0 K and 5 K. This is not the case of Fig. 10a, clear sign that in the depicted emission there are also some particles (ash and/or water droplets)."*

The presence of particles in the plume justify the overestimation of SO2 flux compared to TROPOMI (Sect. 6).

The discussion of viewing geometry and MODTRAN inputs, illustrated in Fig. 11, is a bit confusing. Why are such corrections for wind direction (Eqs. 22 and 23) necessary, when the authors have already corrected for impact of wind direction on "pixel heights" (Eqs. 9 – 12; Fig. 6). What is different about these two sets of corrections?

In Eqs. 22 and 23 the wind direction is not considered. This part of the procedure is related to the way to insert a volcanic plume into the MODTRAN model, as explained at lines 414 etc. The main problem is that with MODTRAN we can only insert a horizontal plume, that is a plume (as shown in fig. 11, named "MODTRAN plume") which occupies an entire and infinite horizontal layer of atmosphere. This means that for MODTRAN the volcanic plume is both over the vent and over the camera. This is not the real case (the only exception is "when the plume moved exactly in the direction of the camera", as written at line 416). Usually in the real case the plume is far from the camera. To make sure that the MODTRAN plume configuration coincides with the real plume configuration (that is the length of the optical paths through and ahead the plume be the same) we must apply Eqs. 22-23 (as written at line 422).

We inserted in the manuscript a new sentence and a "new" Fig. 11, in which we hope that this aspect is more evident:

Line 416
*"This means that, for MODTRAN, the volcanic plume occupies an infinite (in all directions) horizontal layer of atmosphere."*

[Figure]

New Fig.11 inserted in the manuscript

Returning to Fig. 11, do the authors know the "horizontal thickness," or HT, of the plume? This knowledge would require observations of the plume from the side opposite the camera. Do the authors assign an HT canonically? The HT, together with camera inclination ($\theta$) and horizontal distance from plume ($D_0$), defines the length of the optical path through a plume. However, $\theta$ and $D_0$ are measured at the camera and derived from topo maps.

Yes, the reviewer is right, HT is unknown, and, for this reason, it is supposed arbitrarily. Fortunately, as shown in the Sensitivity Study in Sect. 7, the variability of HT is very low sensitive to the SO2 results compared to the other uncertainties of the procedure (see table 2). This aspect is remarked at line 440: *"HT is an input parameter which is usually unknown, but it does not significantly affect the results (see Sect. 7)."*

With HT, $\theta$, and $D_0$ known (or defined canonically), the authors define a "MODTRAN plume" for each line-of-sight (or pixel in the focal plane) and derive estimates of the vertical column density (VCD) of $SO_2$. However, the MODTRAN plumes are segments of the actual plume and, as far as I can tell, the RT modeling does not account for upwelling radiance from the plume beneath the segment in question nor downwelling radiance from the plume above the segment. The camera images suggest that the optical depth of the plumes was significant and, therefore, I suspect that the up- and downwelling radiance was significant.

Yes, the reviewer is right, in this way we neglect the upwelling and downwelling radiance from the lower and upper part of the plume, regarding to a certain optical path. But, on the other hand, MODTRAN considers the presence and the contribution of the plume on the left and on

the right ("MODTRAN configuration"), with respect to a certain optical path. So, there is a sort of partial compensation. By the way, we considered the multiple scattering option in the MODTRAN simulation even if its contribution is quite small. In the figure below you can see the differences between NB temperatures with or without the multiple scattering obtained from MODTRAN for the test case presented. The differences are very low (from 0.1 to 0.5 K).

[Figure]

So, we think that this "error" is not so significant. We inserted a sentence about this aspect in the new manuscript:

Line 423:
*"In the MODTRAN configuration, the model does not account for upwelling and downwelling radiance from the lower and upper part of the plume, regarding a certain optical path. This is partially compensated by the presence of the plume on the left and on the right, regarding a certain optical path. In any case the multiple scattering contribution (used in the MODTRAN simulations) is very low."*

The comparison of VIRSO2 and TROPOMI-based $SO_2$ estimates (Section 6) needs to be scrutinized. The results are described as comparable (e.g., Line 477), but Fig. 14b indicates that the VIRSO2-based estimates were consistently 2X higher than the TROPOMI estimates. The minor overlap between the results is achieved when the errors (which are reported to exceed ±50% for both instruments) are incorporated into the figure.

Line 513 was changed in:
*"Figure 14b shows the SO2 fluxes obtained from the TIR camera and TROPOMI, with the first one greater than about 300-800 t d-1 (tons per day)"*

Actually, the reported error is ±40% for both instruments (not ±50%), as written at line 516:
*"Considering a $\pm 40$ % error for both VIRSO2 (see Sect. 7) and TROPOMI ($\pm 35$ % retrieval error (Corradini et al., 2021) plus $\pm 20$ % for wind speed, quadrature sum)"*

The VIRSO2 images (BB or NB) indicate that the plumes are warmer than the clear-sky temperatures. In other words, the plumes are the sources, or emitters, of the observed radiance. This emission requires that the optical depth of the plume is high enough to minimize the transmission of radiance through the plumes. As the optical depth decreases, due primarily to the dispersion of plumes from their source vents, the plume transmission increases (relative to emission) and the plumes become difficult to detect against the clear-sky background.

The possible scenarios range from perfectly opaque plume to absent plume. In the first case (perfectly opaque plume) the optical thickness of the plume is infinite, the plume transmittance = 0, the plume emissivity = 1, the background radiance transmitted through the plume is zero. In this case the VIRSO2 would measure the real plume temperature. As written at line 494, in the MODTRAN simulations we assumed that the plume is in thermal equilibrium with the surrounding air temperature. So, for an opaque plume, the temperature would be constant and equal to the temperature of the atmospheric layer where the plume is located.

In the second case (transparent or absent plume) the plume optical thickness = 0, the plume transmittance = 1, the plume emissivity = 0 and obviously the VIRSO2 would measure the temperature of the clear-sky only. Between these two extreme cases the plume is semi-transparent, the plume transmittance and emissivity have a value between 0 and 1 and the radiance from the atmosphere behind the plume is able to pass partially through the plume. In this case the VIRSO2 camera measures a temperature warmer of clear-sky temperature but colder of the opaque plume temperature. This is the range where we can apply our procedure.

It is possible to find a detailed description of the different radiance components which contribute to the total radiance measured by the VIRSO2 camera in "Prata, F., Corradini, S., Biondi, R., Guerrieri, L., Merucci, L., Prata, A., and Stelitano, D.: Applications of Ground-Based Infrared Cameras for Remote Sensing of Volcanic Plumes, Geosciences, 14, 82, https://doi.org/10.3390/geosciences14030082, 2024" (see Eqs. 1-3 in Sect. 5).

Here we used directly the MODTRAN simulations, so we think that this description is not necessary (the paper is already quite long as it is).

Conceptually, the TROPOMI detection of $SO_2$ is based on the absorption of back-scattered UV radiance passing through the plumes. Once again, if the optical depth of a plume is too high then the plume becomes the source (through back-scattering) of the radiance. Given the strength of $SO_2$ absorption in the UV, relative to the TIR (Lines 467-469), the plumes detected by VIRSO2 are almost certainly opaque to UV radiance and TROPOMI is measuring back-scattering from the tops of the plumes.

TROPOMI, like all of its predecessors (TOMS, GOME, SCIAMACHY, OMI, GOME-2, OMPS) have been used over the last decades for the monitoring of anthropogenic and volcanic SO2 emissions

worldwide. It is certainly true that for very large SO2 columns and high ash particles content, such as occurring during strong explosive volcanic eruptions, the DOAS technique typically underestimates the SO2 VCD. But this is not the situation that we have for the test case used in the manuscript, in which a normal degassing plume is present.

Furthermore, as written at line 508, the TROPOMI SO2 flux is computed by applying the traverse method, that is, as written at lines 509-512, *"all the transects perpendicular to the volcanic cloud axis are considered"* and *"this makes it possible to obtain an SO2 flux trend for several hours before the image acquisition"*. In this case the camera measurements are at 5-7 UTC in the morning, while the TROPOMI pass is at 11:33 UTC. Considering a wind speed of 2 m/s and a time delay of 5.5 hours, this means that the volcanic cloud portion detected by TROPOMI and used to compute the SO2 flux between 5-7 UTC is located at about 40 km far from the volcano in the 11:33 UTC satellite image. So, it is reasonable to suppose that the natural dispersion of the volcanic cloud (in a normal degassing situation), makes it not opaque for TROPOMI.

Putting aside my concerns about the derivation of $T_{NB}$", my interpretation of Fig. 14b is based on the effects of optical depth. The TROPOMI measurements did not sample the interiors of the UV-opaque plumes and, as a result, the VCD were under-estimated. The VIRSO2 measurements did sample the interiors of the plumes, and the resulting estimates of VCD were higher than the TROPOMI estimates. Given the significant role of optical depth, it is hard to think of a scenario where the direct comparison of VIRSO2 and TROPOMI estimates is possible.

As noted above, the plumes were the sources of the TIR radiance and, because the radiance did not pass through the entire plume, it is likely that the VIRSO2 procedure under-estimated VCD.

By considering the MODTRAN simulations, given that the $DT_{NB}$ ranges between about 0-23 K (fig. 10b, reported below), the plume is not completely opaque because in this temperature range the simulated $DT_{NB}$ values vs the SO2 contents (shown in fig. 12b and reported below) are not horizontal but still increasing. This means that the background radiance passes partially through the plume and the VIRSO2 is capable of detecting the entire plume. From MODTRAN simulations (fig. 12b) the plume opacity occurs for example for elevation angle = 10 deg and SO2 > 20 g/m2 (the line starts to become almost horizontal).

[Figure]

Fig. 10b of the manuscript

Fig. 12b of the manuscript

So, for this test case, we remark our final considerations, as reported at line 518: *"In this case, as written in Sect. 5, the overestimation of VIRSO2 is probably due to the presence of water vapor particles condensed in the volcanic plume."*, not considered in the MODTRAN simulations in this work.

---

## Author Comment (AC2)

Review of "A Novel Simplified Ground-Based TIR System for Volcanic Plume Geometry, SO2 Columnar Abundance, and Flux Retrievals"

We would like to sincerely thank the reviewer for his work and the time spent reading our article. His suggestions have helped to significantly clarify some parts of the paper.
The line numbers in the authors' comments refer to the new manuscript revised according to the reviewers' suggestions.

This paper presents a comprehensive methodology for retrieving SO2 column density using a ground-based TIR system (2 IR cameras and one visible camera). The study is well-structured, detailing the three key steps: computing the instrument view geometry and the preparation of the data from both IR cameras (one broad band and one narrow band with a filter at 8.7µm), calibration (using the radiative transfer model called MODTRAN), and SO2 retrieval (using look-up tables from MODTRAN). The authors provide an analysis of error propagation and uncertainty quantification, making the findings valuable for the atmospheric and volcanology communities. Overall, this is a well-executed study that presents a useful methodology for SO2 flux retrievals, with a balanced discussion of its advantages and limitations.

**Suggestions of technical corrections:**

Line 37: "but are punctual" → "but are limited to specific locations"

OK, thanks, done.
Lines 50-51: "and as part of a continuous and real-time volcanic monitoring system." → ", and as part of a continuous, real-time volcanic monitoring system."

OK, thanks, done.
Lines 77-78: "manufacturer supplied" → "manufacturer-supplied"

OK, thanks, done.
Line 80: Why is the conversion into brightness temperature performed at 10.02 µm?

Thanks for the question. MODTRAN simulations were performed at 1 cm-1 step and the output radiances are in mW m-2 sr-1 cm-1 units. The Broadband squared Spectral Response Function is considered equal to 1 between 1282 cm-1 (7.8 µm) and 714 cm-1 (14 µm). So, the average wavelength is 998 cm-1 which corresponds to 10.02 µm. We have added some details in the text to better clarify this aspect:

Line 77: "*As the figure shows, a simple top-hat function between 1282 cm-1 (7.8 µm) and 714 cm-1 (14 µm) was considered for BB since there is no specific information about it from SEEK Thermal. For NB, the manufacturer-supplied spectral transmittance (normalised to 1) of the 8.7 µm filter was used. The MODTRAN spectral radiances (in mW m-2 sr-1 cm-1 units and at 1 cm-1 step) were weighted by the two SRFs and then converted into brightness temperatures by inverting the Planck function considering a central wavelength of 998 cm-1 (10.02 µm) and 1151 cm-1 (8.69 µm) respectively.*"

Figure 2a: The quality of the text box could be improved.

OK, thanks, done.

Line 145: In the legend of Figure 4, replace "m2" with "m²".

OK, thanks, done.

Lines 162-166: Equations 9, 10, 11, and 12 are not easy to conceptualise. To help the reader, a new illustration could be added to clearly show x(j) and explain mx and qx. Alternatively, additional text could be included to clarify these equations.

OK, we have modified fig. 5. We hope that now the equations are more clear.

[Figure]

New Fig. 5 inserted in the manuscript

Line 169: In Equation 3.8, does the reference correspond to Equation 8?

Thanks. The reference is to Equations from 3 to 8 so we changed the texts in "Eqs. 3–8" (en dashed has to be used to indicate a range).

General question regarding Section 3.1: How is the wind direction (ω) determined? Is it estimated during the field campaign, or is it derived from GEO (SEVIRI) or LEO (TROPOMI) data? If this is the case, it should be explicitly stated.

As written in lines 161-162, ω is not the wind direction but it's the relative azimuth angle between the wind direction and the focal plane of the camera. In this case the wind direction was taken from an atmospheric forecasting model from ARPA ("Agenzia Regionale Prevenzione Ambiente"). Based on the wind direction and the camera position and

orientation, ω = 26° from Piano del Vescovo while $\omega = 52°$ from Nicolosi. We added this sentence in the manuscript:

Line 179: *"Wind data was taken from the mesoscale model of the hydrometeorological service of Agenzia Regionale per la Protezione Ambientale (ARPA) Emilia Romagna, which is named ARPASIM (Scollo et al., 2009), and considering an hourly model output from 72-h weather forecast provided for Etna every 3 h."*

Lines 180-184: SEVIRI retrievals estimate a plume top altitude of 6.9 km at 00:55 UTC on April 1, 2021. You state that this is in good agreement with the VIRSO2 nighttime measurements. Could you specify the volcanic plume top altitude obtained using VIRSO2's field of view and geometric considerations, both with and without wind correction?

The comparison of the plume height from SEVIRI and from the VIRSO2 camera here is just qualitative. The purpose of this part is to describe the method and show the importance of considering the effect of the wind. An accurate quantitative comparison would require that the two measurements (from satellite and from ground) were taken at the same instant, but unfortunately, we started collecting measurements from Piano del Vescovo at 02:29 UTC. Anyway, the fig. 6b clearly shows that for this image (02:29 UTC) the top plume height is a bit lower than 7 km a.s.l. while in the fig. 6a (without wind correction) the top plume height is about 8.5 km asl (as written at line 184).

Lines 190-193: The error increases for ω > 45°. Could you provide an explanation for this? Additionally, what conclusions can be drawn from the comparison between this study and the tool provided by Snee et al. (2023)?

As written at line 161, ω ranges between -90° and +90°. These extreme values (ω=90° or ω=-90°) mean that the wind direction is parallel to the central line of sight of the camera $D_0$, that is the plume goes perpendicular to the focal plane. In this case it's impossible to apply the wind correction method (in the Eq. 9 the tangent becomes infinite). Apart from these two specific cases, the increasing of |ω| produces an increased sensitivity on wind direction and camera orientation. This means that small errors in these quantities can result in large errors in the wind-corrected height.

The comparison with the tool provided by Snee et al. (2023) wants to demonstrate the accuracy of our simple method for a relatively large number of cases. The calibration tool of Snee et al. (2023) uses a more complex and more accurate set of formulas, and it has already been published. The conclusions of this comparison are that if for |ω|<45° our simple method presented has a very low error.

We inserted a comment in the new manuscript (line 200):

*"So, during field measurements, it would be desirable to position the VIRSO2 so that the camera's focal plane is parallel to the wind direction or tilted at an azimuth angle of no more than 45 degrees."*

Lines 200-202: The text mentions "the next section," but the order of presentation is Sections 4.1, 4.3, and 4.2. Consider reordering the text to follow the sequence 4.1, 4.2, and 4.3, or, if this is not logical, switching the order of Sections 4.3 and 4.2.

OK, we changed the order of these sentences. The order of the Sections is logical and follows the steps of the calibration procedure (see the flowchart fig. 2a).

Line 208: "*Here different effects must be taken into account: at first, the non-perfect transmissivity of the 8.7 µm filter produces a "ghost image". Then, an adjustment is important also for the BB camera, considering that the clear sky temperature often doesn't match with the MODTRAN simulations. Finally, the filter temperature affects strongly the NB measurements, so a calibration of this band is necessary."*

Lines 200-202: "at first, the non-perfect transmissivity of the 8.7 µm filter produces a 'ghost image.' Then, the filter temperature affects the NB measurements, and finally, an adjustment is necessary for the BB camera, considering that the clear sky temperature often does not match the MODTRAN simulations." The mention of "non-perfect transmissivity" and "filter temperature effects" for the NB measurements appears closely related. Additionally, some introductory information about the use of MODTRAN would be beneficial. Improving the clarity of these three lines would help the reader grasp the concept more easily.

The order of these sentences is changed as suggested by the reviewer. Surely, the "non-perfect transmissivity" and the "temperature effects" of the NB filter are related but as shown in the following sections (4.1 and 4.3), these two problems are addressed and resolved separately. The use of MODTRAN (radiative transfer model) to obtain the quantitative retrieval of SO2 was already mentioned in the abstract and in the introduction. Then its use is deeply described in the following sections (4.2, 4.3, 5).

Line 206: The phrase "do not frame exactly the same scene" describes a common issue known as an X/Y shift between the two cameras. This terminology could be included for clarity.

OK, thanks, done.

Line 213: "For the BB camera, this effect is small enough that a correction is not required (Prata et al., 2024)." Given that the vignetting effect results in a range of 4 K for the NB and only 0.4 K for the BB, it seems reasonable to assume that vignetting correction is not necessary for the BB camera.

Yes, exactly, the reviewer is right.

Lines 230-232: "A 'correction image' (Fig. 7c) is simply obtained as the difference between the black target temperature image Tn,NB(i, j) and its mean value MEAN(Tb,NB) over the 320 × 240 array." It is unclear why MEAN(Tb,NB) is subtracted from Tn,NB(i, j). Could you provide an explanation? Is this a way to normalise the ghosting effect? What is the purpose of applying this offset? Clarifying this would be helpful.

When a "black target" measurement is made (that is a measurement with a purpose -built black target placed in front of the camera), we obtain from NB camera an image containing only the "ghost image" (see the figure below).

[Figure]

Assuming that the black target has a uniform temperature (due to the high proximity of the sensor to the target this seems a correct assumption), to remove the "ghost" effect from this image it is enough to subtract the image itself with the real temperature of the black target. But, as written at line 231, given that this correction is only of a "geometric" type (at this step it doesn't matter if the resulting image is correct in temperature, because a temperature calibration of the NB is performed later, sect. 4.3), the actual temperature of the plate is not required. So here it is enough to subtract the image itself with a single temperature value, which can be for example the minimum value or the mean one. To clarify this important aspect, we inserted a new comment in the manuscript:

Line 253: *"It's important to remark that at this step it doesn't matter if the resulting image $T'_{NB}$ is correct in temperature, because the temperature calibration of the NB is performed later (Sect. 4.3); so, for example, using the minimum value of the black target image instead of the mean one, it wouldn't change the final result, that is the SO2 content."*

If the "ghost" effect in the black target scene was exactly the same as the one in the scene with the volcanic plume, the "ghost" image would be perfectly removed. This usually doesn't happen, because the ghost image depends to both the different temperatures of collected scene, as well as to any variations in the internal temperature of the instrument. For this reason, this simple geometric correction does not always give optimal results; in some cases, the "ghost" effect in the volcanic plume image is only partially removed (lines 256-260).

Line 287: In Equation 15, it appears that the term B is not explicitly defined.

B is the Planck function, we added this information in the text.

Line 306: In Figures 8a and 8b, using the same Y-axis range (e.g., 210–290 K) could improve visual consistency.

OK, thanks, done.

Line 313: In Section 4.3, you wrote, "as described in Sect. 4." Did you mean "As described in the introduction of Section 4"?

Yes, thanks, done.

Lines 322 & 331: To fully understand Equations 16 and 17, it is crucial to define the term B.

OK, thanks, done.

Line 351: In Section 4.3, you wrote, "as already reported in Sect. 4." Please specify whether this refers to the introduction of Section 4, Section 4.1, or Section 4.2.

OK, it's the introduction, done.

Lines 350-355: It would be helpful to explicitly define $T_{BB}$(sky), $T_{BB}$(ground), $T_{NB}$(sky), and $T'_{NB}$(ground).

OK, thanks, we have added in the text at Line 368: *"The labels (sky) and (ground) represent the average temperatures of two small areas of the image related to clear sky and ground respectively (Fig. 9a)."*

Line 374: In Figure 10a, there still appears to be some vignetting effect. If confirmed, this should be mentioned.

OK, thanks, we have added in the text at Line 402: *"As already noted in Fig. 8a, some small border effects (vignetting) are present in the BB image, so we obtained some $DT_{BB}$ values > 0 which are not related to the plume's presence."*

Line 401: Equation 22 does not seem to match Figure 11. The equation states:

BH = $D_0$ tan(θ)

However, if $B_{MP}$ represents the mean altitude of the MODTRAN plume layer above sea level, the correct expression should be:

$B_{MP}$ = ($D_0$ + HT/2) tan(θ) + $h_0$

$B_{MP}$ should also be added to Figure 11.

OK thanks, there was a small error in fig. 11 that now we have corrected ($D_0$ was wrongly indicated). The $D_0$ distance (perpendicular distance between the camera and the image plane) is computed considering the geographical position (Lat; Lon) of the camera and the vent and the column position "*jc*" of the vent in the VIRSO2 image (the angle $\varphi(jc)$ is known). So, we think that Eq. 22 which permits to compute the bottom altitude of the plume is correct as it is:

$$BH = \left(D_0 - \frac{HT}{2}\right) \cdot tan\,(\theta)\, + h_0$$

In the MODTRAN model we must insert the top and the bottom of the plume altitude. The BH and VT parameters (Eqs. 22-23) give us this information. The mean altitude of the plume (as suggested by the reviewer) is not necessary.

[Figure]

New Fig.11 inserted in the manuscript

Line 415: It may be useful to remind the reader that $T_s$ represents the clear sky temperature.

Sorry, here there was a typo in the text. We have added the missing sentence at Line 445: *"As mentioned in Sect 4.2, the MODTRAN LUTs were recomputed with a more appropriate sky temperature, in this case: Ts = 226 K."*

Line 429: In Figure 13a, there still appears to be some vignetting effect. If confirmed, this should be mentioned.

OK thanks, we added at Line 456: *"In the upper part of the image, some edge effects still remain. These pixels must be excluded in the computation of SO2 mass and flux (Sect. 6)."*